# Bacterial flagellar capping proteins adopt diverse oligomeric states

**Sandra Postel[1], Daniel Deredge[2], Daniel A Bonsor[1], Xiong Yu[3], Kay Diederichs[4], Saskia Helmsing[5], Aviv Vromen[6], Assaf Friedler[6], Michael Hust[5], Edward H Egelman[3], Dorothy Beckett[7], Patrick L Wintrode[2], Eric J Sundberg[1,8,9]***

[1]Institute of Human Virology, University of Maryland School of Medicine, Baltimore, United States; [2]Department of Pharmaceutical Sciences, University of Maryland School of Pharmacy, Baltimore, United States; [3]Department of Biochemistry and Molecular Genetics, University of Virginia, Charlottesville, United States; [4]Department of Biology, University of Konstanz, Konstanz, Germany; [5]Department of Biotechnology, Institute of Biochemistry, Biotechnology and Bioinformatics, Technische Universität Braunschweig, Braunschweig, Germany; [6]Institute of Chemistry, The Hebrew University of Jerusalem, Jerusalem, Israel; [7]Department of Chemistry and Biochemistry, University of Maryland College Park, Baltimore, United States; [8]Department of Medicine, University of Maryland School of Medicine, Baltimore, United States; [9]Department of Microbiology and Immunology, University of Maryland School of Medicine, Baltimore, United States

**\*For correspondence:**
esundberg@ihv.umaryland.edu

**Abstract** Flagella are crucial for bacterial motility and pathogenesis. The flagellar capping protein (FliD) regulates filament assembly by chaperoning and sorting flagellin (FliC) proteins after they traverse the hollow filament and exit the growing flagellum tip. In the absence of FliD, flagella are not formed, resulting in impaired motility and infectivity. Here, we report the 2.2 Å resolution X-ray crystal structure of FliD from *Pseudomonas aeruginosa*, the first high-resolution structure of any FliD protein from any bacterium. Using this evidence in combination with a multitude of biophysical and functional analyses, we find that *Pseudomonas* FliD exhibits unexpected structural similarity to other flagellar proteins at the domain level, adopts a unique hexameric oligomeric state, and depends on flexible determinants for oligomerization. Considering that the flagellin filaments on which FliD oligomers are affixed vary in protofilament number between bacteria, our results suggest that FliD oligomer stoichiometries vary across bacteria to complement their filament assemblies.

## Introduction

Pathogenic bacteria cause a multitude of deadly human diseases. Many of these microbes possess flagella, molecular machines responsible for cell motility, adherence to host cells, and pathogenicity (*Duan et al., 2013*; *Haiko and Westerlund-Wikstrom, 2013*). Flagella are helix-shaped hollow attachments formed predominantly by thousands of copies of the protein flagellin (also called FliC), anchored in the bacterial membrane by a hook (or joint) that is attached to the basal body and that is composed of rotary motor proteins (*Arora et al., 1998*). A proton motive force typically drives the propeller motion of flagella (*Berg, 2003*), resulting in swimming motility. A FliD (also called HAP2) oligomer forms the cap protein complex that is located at the tip of the flagellar filament (*Yonekura et al., 2000*). This complex controls the distal growth of the filament by regulating the

**eLife digest** Many bacteria, including several that cause diseases in people, have long whip-like appendages called flagella that extend well beyond their cell walls. Flagella can rotate and propel the bacteria through liquids, such as water or blood, and they are constructed primarily from thousands of copies of a single protein called flagellin. When flagella are built, the flagellin proteins are placed in their proper positions by another protein called FliD, several copies of which form a cap on the end of flagella. Without FliD, bacteria cannot properly assemble flagella and, thus, can no longer swim; this also hinders their ability to cause disease.

Determining the three-dimensional structure of a protein, down to the level of its individual atoms, can provide unique insights into how the protein operates. However, no one had resolved the structure of a FliD protein from any bacterium to this level of detail before.

Now, Postel et al. report the high-resolution structure of a large fragment of FliD from the bacterium *Pseudomonas aeruginosa*. The structure reveals that parts of this FliD protein are shaped like parts of other proteins from which flagella are constructed, including the flagellin protein that FliD places into position. Some parts of the FliD protein are also very flexible and these parts of the protein are responsible for holding numerous FliD proteins together as a cap. Finally, Postel et al. saw that six copies of FliD bind to one another to form a protein complex on the end of flagella. This last finding was particularly unexpected since it was thought that all FliD proteins formed five-membered cap complexes, an assumption that was based largely on studies of FliD from another bacterium called *Salmonella*.

The current structure covers about half of the FliD protein, and so the next challenge is to determine the structure of the full-length protein. An improved understanding of the structure of FliD may, in future, help researchers to design drugs that stop bacteria from building flagella and, therefore, from swimming and causing disease.

assembly of FliC molecules, which are transported through the hollow filament from the cytoplasm to the tip of the flagellum.

The dynamic movement of FliD in this assembly was modeled based on low (~26 Å)-resolution cryo-electron microscopic (EM) structures of the *Salmonella* serovar Typhimurium flagellum-cap complex (*Maki-Yonekura et al., 2003*; *Yonekura et al., 2000*, *2003*), which adopts the shape of a five-legged stool with flexible leg domains that regulate the assembly of new FliC molecules onto the tip of the growing flagellum (*Maki-Yonekura et al., 2003*). It has been suggested that the plate of the stool is formed by core regions of the FliD molecule, and that disordered/flexible regions form the five leg structures (*Vonderviszt et al., 1998*) that are known to interact with the FliC filament. FliD exhibits low sequence similarity to the flagellar hook proteins and to FliC. Nevertheless, it shares the disordered terminal regions of these flagellar proteins, a common structural characteristic that is thought to control the polymerization of flagellar proteins and to play an important role in interaction with the FliC filament (*Vonderviszt et al., 1998*). These regions are the most conserved in FliD sequences across bacteria. Flagellum-mediated motility is crucial for the virulence and pathogenicity of numerous bacteria, including *Campylobacter jejuni* (*Black et al., 1988*), *Salmonella* (*Allen-Vercoe and Woodward, 1999*; *Marchetti et al., 2004*), *Escherichia coli* (*La Ragione et al., 2000*), *Vibrio cholera* (*Krukonis and DiRita, 2003*), and *Pseudomonas aeruginosa* (*Arora et al., 2005*), as well as the major causative agent of gastric cancer *Helicobacter pylori* (*Kim et al., 1999*). To date, however, no high-resolution structure of any FliD protein exists. To better define the roles of FliD in bacterial motility and pathogenesis, we determined the first X-ray crystal structure of FliD at 2.2 Å resolution, and assessed the structural contributions of its flexible regions using a multitude of complementary biophysical and functional analyses.

# Results

## Crystal structure of the FliD protein from *P. aeruginosa* PAO1

To facilitate crystallization of FliD from the *P. aeruginosa* PAO1 strain, we deleted the predicted coiled-coil domains on both the N- and C-termini of full length FliD, which has 474 residues (FliD$_{1-474}$), to generate the truncated FliD$_{78-405}$ (*Figure 1a*, *Figure 1—source data 1*). We expressed FliD$_{78-405}$ in *E. coli* with an N-terminal His$_6$-tag and purified it to homogeneity by Ni$^{2+}$-NTA, size exclusion and anion exchange chromatography. We improved initially weakly diffracting crystals of FliD$_{78-405}$ by random matrix microseed screening (*Bergfors, 2003*), yielding crystals that diffracted to 2.2 Å resolution. In the absence of any homologous protein that could be used as a model for molecular replacement, we crystallized a seleno-methionine derivative of FliD$_{78-405}$ that included four leucine-to-methionine mutations (FliD$_{78-405}$/L$_4$–M$_4$). This crystal provided phase information sufficient to build an initial model, which we used subsequently for molecular replacement with the native FliD$_{78-405}$ dataset (*Figure 1—source data 1*). We modeled residues 80–273 into clear electron density, including all side chains, but observed density of increasingly poor quality in the C-terminus beyond residue 273 (*Figure 1—figure supplement 2a*). Thus, we were able to model with

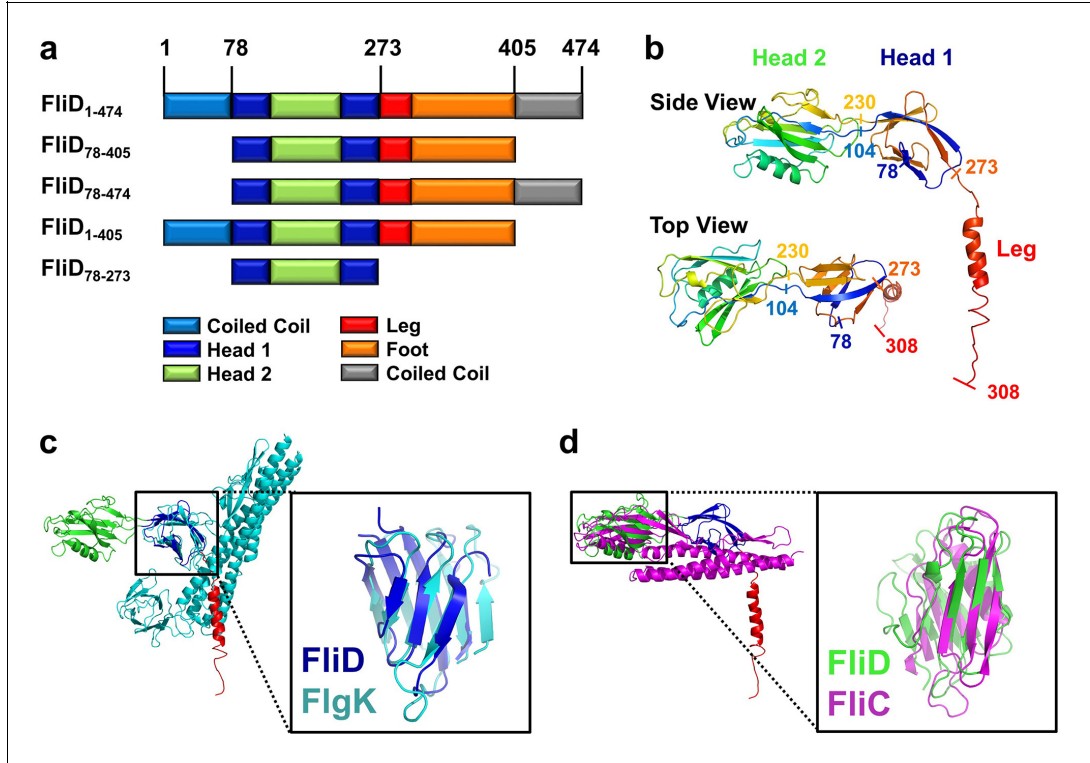

**Figure 1.** Crystal structure of *Pseudomonas* FliD reveals structural similarity to other flagellar proteins. (a) Schematic representation of the FliD proteins used in these studies. Protein domain/region boundaries are labeled and are drawn approximately to scale. (b) Crystal structure of the *Pseudomonas* FliD$_{78-405}$ monomer subunit with spectrum coloring from the N-terminus (blue) to the C-terminus (red). Head domain 1, head domain 2 and the leg region are indicated. (c) Superposition of the FliD$_{78-405}$ crystal structure (domain coloring as in panel (a)) and *Burkholderia* FlgK/HAP1/hook filament capping protein (cyan). (d) Superposition of the FliD$_{78-405}$ crystal structure (domain coloring as in panel (a)) and *Pseudomonas* flagellin/FliC (magenta).

The following source data and figure supplements are available for figure 1:

**Source data 1.** Crystallographic data collection, phasing and refinement statistics.

**Figure supplement 1.** Protein sequence of FliD$_{1-474}$.

**Figure supplement 2.** Electron density and protein degradation of FliD crystals.

confidence only a single α helix in this region, corresponding to residues 274–308, with incomplete side chain structures. To determine whether the remaining region of the protein actually existed in the crystals and not just in the protein preparation used for crystallization, we analyzed crystals using liquid chromatography-mass spectrometry (LC-MS) and SDS-PAGE. Both analyses indicated that the crystals consisted of an approximate 50:50 mixture of the $FliD_{78-405}$ protein used for crystallization and a further proteolyzed version with a molecular weight of about 27 kDa. The N-terminal $His_6$-tag is still detectable by Western blot (*Figure 1—figure supplement 2b*). Thus, the proteolyzed form corresponds approximately to residues 78–319 of FliD. The 86 residues absent from the C-terminus in a population of FliD proteins are clearly not required for crystal packing, suggesting that they are highly flexible even in a crystalline environment.

## FliD is structurally similar on the domain level to FliC and FlgK

Our crystal structure of $FliD_{78-405}$ reveals that it consists of two discreet regions with distinct conformational properties, corresponding to a stable head region and a flexible and/or disordered leg region (*Figure 1b*). The head region is itself comprised of two separate, but entwined, protein domains. Residues 80–101 form two β strands that belong to the first domain (Head 1), the second domain (Head 2) is formed in its entirety by the contiguous residues 104–230, and residues 231–273 then complete the first domain. The second domain is, thus, a loop insertion of the first domain. We searched for structural homologs of these domains in the Protein Data Bank and found that the first domain of the head region (Head 1) exhibits high structural similarity (RMSD=2.5 Å), despite low sequence identity (14%), to the FlgK/HAP1/hook filament capping protein of *Burkholderia pseudomallei* (PDB code 4UT1; *Figure 1c*). Likewise, the second head region domain (Head 2) exhibits high structural similarity (RMSD=2.7 Å), despite low sequence identity (17%), to the FliC/flagellin protein of *P. aeruginosa* (PDB code 4NX9; *Figure 1d*). In contrast to the head region, the leg region of FliD is highly flexible, as indicated by the paucity of electron density corresponding to residues 274–405 (*Figure 1—figure supplement 2a*). Despite this, we were able to model the initial α helical structural element, corresponding to residues 274–308, of this region. This helix extends from the axis of the head region at a nearly perpendicular angle, resulting in an L-shaped monomer subunit structure (*Figure 1b*).

## FliD from *P. aeruginosa* PAO1 forms a hexamer

In the crystal, $FliD_{78-405}$ monomer subunits are arranged in hexamers, resulting in a shape akin to a six-pointed star when viewed from the top of the FliD oligomer (*Figure 2a*), which corresponds to the distal end of the growing flagellum. This star shape has a minimum inner diameter of 48 Å and a maximal outer diameter of 136 Å. When viewed from the side (*Figure 2b*), the FliD hexamer appears as a six-legged stool, the legs of which extend 55 Å below the bottom of the head region. Additional crystallographic symmetry results in the stacking of hexamers in alternating head-to-head and leg-to-leg orientations (*Figure 2c*). The leg-to-leg stacking forms dodecamers, resulting from the helix–helix interaction of the residues 274–302 of stacked molecules and the interaction of residues 303–308 of one $FliD_{78-405}$ molecule with Head 1 domain of a stacked molecule, burying a surface area of 1362 $Å^2$. The formation of dodecamers may be unique to the $FliD_{78-405}$ fragment, as this strand could potentially be replaced by additional N-terminal residues in the full-length $FliD_{1-474}$ protein. All of the morphologies observed for *Pseudomonas* FliD are highly reminiscent of the pentamer/decamer oligomeric organization of *Salmonella* FliD as determined by low-resolution cryo-EM analysis (*Maki-Yonekura et al., 2003*) (*Figure 2d*). Indeed, despite the difference in the stoichiometries of the *Pseudomonas* and *Salmonella* FliD oligomers, the gross measurements are nearly identical for these two proteins of similar molecular weight. In our crystal structure, *Pseudomonas* FliD measures 136 Å in diameter with a head region that is 30 Å deep and a leg region that is 55 Å long; whereas, in the cryo-EM structure, *Salmonella* FliD measures 145 Å in diameter with a head region that is 30 Å deep and a leg region that is 55 Å long (*Maki-Yonekura et al., 2003*).

The stoichiometry of *Pseudomonas* FliD differs from that of *Salmonella* FliD and our *Pseudomonas* FliD crystals belong to the *P6* space group, which could possibly force a non-physiological oligomeric organization of subunits. Thus, we confirmed that the hexameric assembly of $FliD_{78-405}$ occurs not only in the crystalline environment but also in solution using negative stain EM class averaging

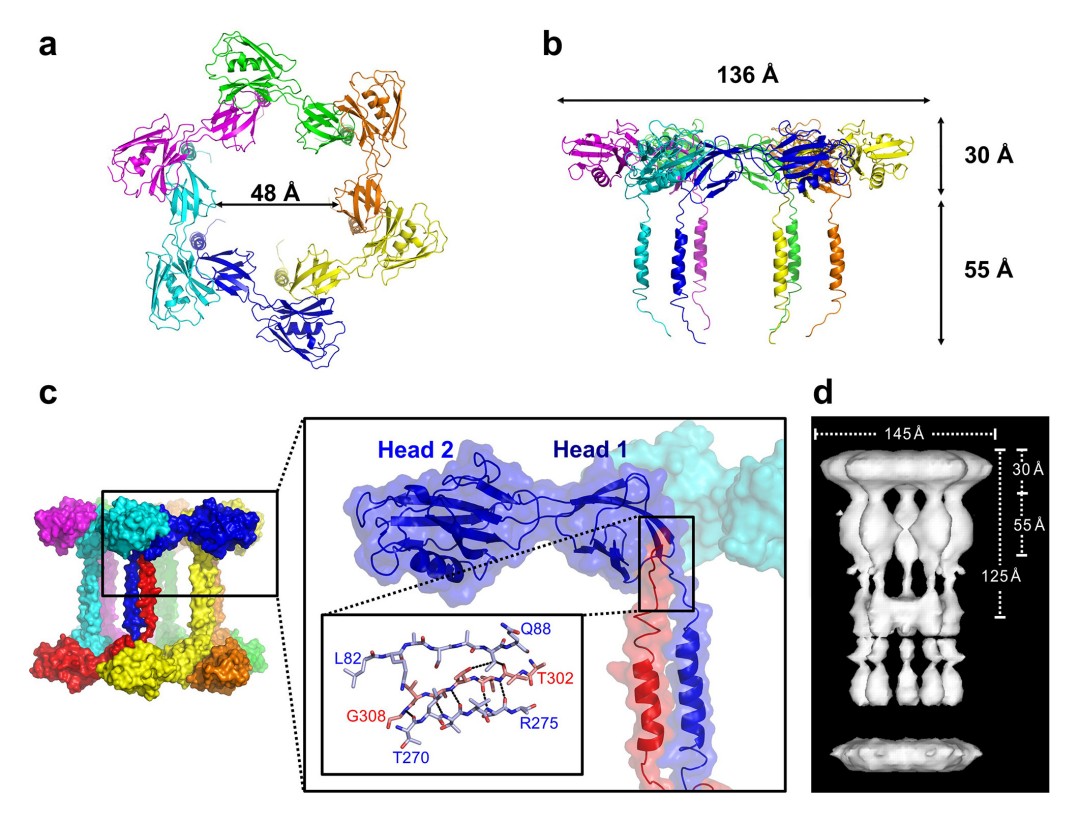

**Figure 2.** *Pseudomonas* FliD forms hexamers in crystals. (**a**) Top view, cartoon representation of the FliD$_{78-405}$ hexamer. Each monomer subunit is colored distinctly and inner diameter dimension is indicated. (**b**) Side view, cartoon representation of the FliD$_{78-405}$ hexamer. Each monomer subunit is colored distinctly. Outer dimensions are indicated. (**c**) FliD$_{78-405}$ hexamers as arranged in the crystal are stacked head-to-head and leg-to-leg (shown) in an alternating fashion, with residues 303–308 assembling in the Head 1 domain of an opposing molecule (close-up views) leading to a dodecameric crystal packing. (**d**) Cryo-EM structure of *Salmonella* FliD from (*Maki-Yonekura et al., 2003*) for comparison.

(*Tang et al., 2007*), resulting in an image that clearly exhibits a hexameric assembly (*Figure 3a*). These data are consistent with the oligomerization state that we detected in the crystal structure, in which a view looking down the α-helices of the leg reveals a six-membered ring organization. We also verified that both FliD$_{78-405}$ and full-length FliD$_{1-474}$ form oligomers by both analytical ultracentrifugation (AUC) and crosslinking analyses. We found that the FliD$_{78-405}$ fragment that we crystallized oligomerizes up to a dodecameric state (*Figure 3b,c*), similar to the crystallographic assembly (*Figure 2c*). Additionally, we collected small-angle X-ray scattering (SAXS) data of FliD$_{78-405}$, for which the calculated radial distribution function (*Figure 3d*) is characteristic of an oligomeric assembly forming a hollow sphere (*Svergun and Koch, 2003*). These data produce a molecular envelope that superimposes well with our dodecameric FliD$_{78-405}$ X-ray crystal structure (*Figure 3d*). Full-length FliD$_{1-474}$, by contrast, forms up to hexamers in solution (*Figure 3e,f*), which are likely to represent the physiologically relevant oligomerization state of this protein on the tip of the flagellum. The predominant tetrameric species in solution identified by AUC may be a stable intermediate on the path to hexamer formation (*Figure 3e*).

As the hexameric assembly of full-length *Pseudomonas* FliD$_{1-474}$ proved to be unstable in solution in the absence of the flagellar filament, we sought to stabilize it using our newfound understanding of its structure. We used our crystal structure of FliD$_{78-405}$, as input to Disulfide by Design 2.0 (*Craig and Dombkowski, 2013*), to identify cysteine mutations that would lead to a stable, disulfide-bridged hexameric FliD$_{1-474}$. We found that when two residues within neighboring head domain subunits, I167 and D253, were each mutated to a cysteine residue (*Figure 4a*) a stable, hexameric full-length FliD$_{1-474(I167C/D253C)}$ resulted under non-reducing conditions, as shown by SDS-PAGE

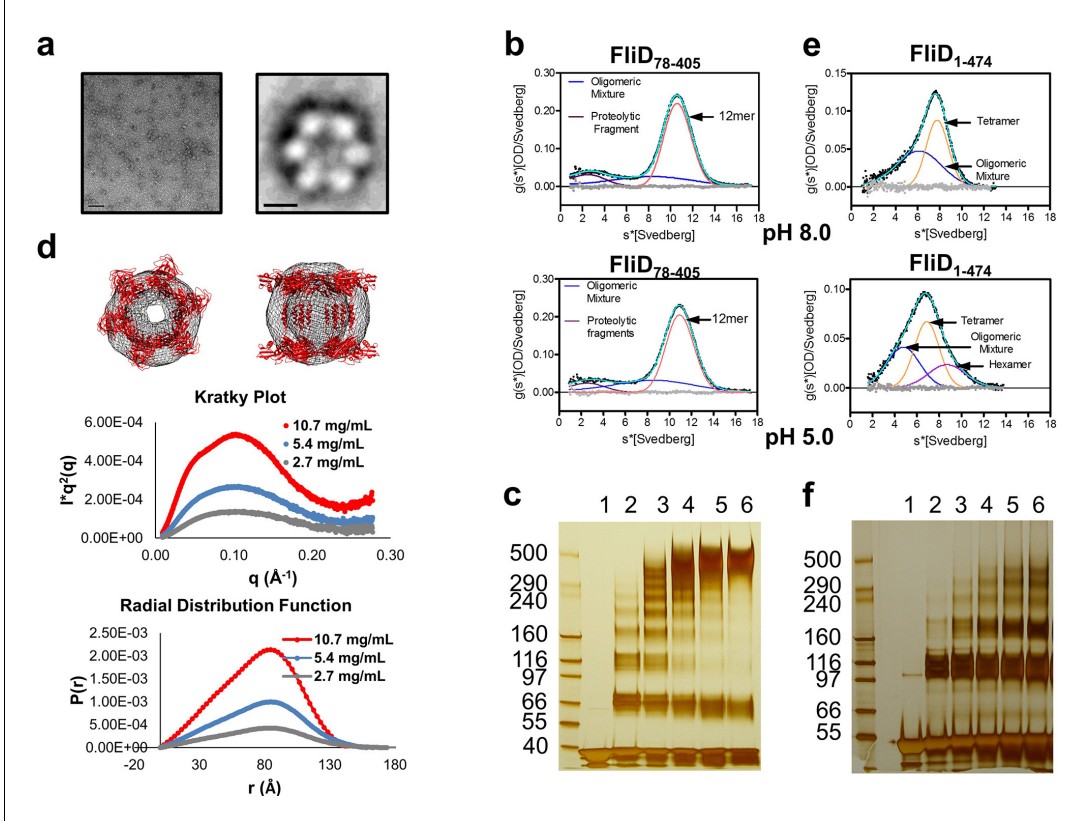

**Figure 3.** *Pseudomonas* FliD oligomerization. (a) Negative stain EM image of FliD$_{78-405}$: left, single particles (scale bar=1000 Å); right, class-averaged particle (scale bar=50 Å). (b) AUC analysis of FliD$_{78-405}$ proteins at pH 8.0 (upper panel) and pH 5.0 (lowel panel) indicates that it forms dodecamers in solution. (c) Silver-stained SDS-PAGE after chemical crosslinking of FliD$_{78-405}$. (d) SAXS analysis of FliD$_{78-405}$. Kratky plot (*I*\*q$^2$ *versus q*) and radial distribution function calculated by GNOM, and SAXS envelopes calculated by DAMMIF, with superimposed crystal structures are shown for FliD$_{78-405}$ at 10.7 mg/mL (red, used to calculate envelope), 5.4 mg/mL (blue) and 2.7 mg/mL (grey). (e) AUC analysis of full length FliD$_{1-474}$ proteins at pH 8.0 (upper panel) and pH 5.0 (lower panel) indicates a mixture of oligomers, including tetramers and hexamers. (f) Silver-stained SDS-PAGE after chemical crosslinking of FliD$_{1-474}$.

(*Figure 4b*) and SAXS (*Figure 4c*) analyses. We also confirmed that the cysteine bridges resulted in the expected interfaces between head domains by employing mass spectrometry to compare the peptide coverage under reducing and non-reducing conditions (*Figure 4—figure supplement 1a*) and by successfully detecting the correct cysteine bridges (C167–C253) while ruling out non-specific cysteine bridging (C167–C167 and C253–C253) (*Figure 4—figure supplement 1b–e*).

To show that the hexameric form of *Pseudomonas* FliD is functional in vivo, we complemented the *fliD* transposon strain PW2975 (Δ*fliD*) with wildtype *fliD$_{1-474}$* and hexamer-stabilized *fliD$_{1-474}$* *(I167C/D253C)*, resulting in *Pseudomonas* PAO1 strains Δ*fliD*/*fliD$_{1-474}$* and Δ*fliD*/*fliD$_{1-474(I167C/D253C)}$*, respectively. We found that swimming motility that was lost in the Δ*fliD* strain was restored in both Δ*fliD*/*fliD$_{1-474}$* and Δ*fliD*/*fliD$_{1-474(I167C/D253C)}$* complementation strains, similar to our observations in the wildtype *Pseudomonas* PAO1 strain (*Figure 4d*). Using antibodies that we generated by phage display to *Pseudomonas* PAO1 FliD, we confirmed the expression of full-length FliD proteins by Western blot analysis from preparations of flagella isolated from live bacteria from both Δ*fliD*/*fliD$_{1-474}$* and Δ*fliD*/*fliD$_{1-474(I167C/D253C)}$* complementation strains (*Figure 4e*). In flagella preparations from the Δ*fliD*/*fliD$_{1-474(I167C/D253C)}$* complementation strain, the FliD$_{1-474(I167C/D253C)}$ protein produced by *Pseudomonas* maintains its hexameric oligomeric state (*Figure 4e*). In contrast, subsequent to flagella isolation from live bacteria, FliD$_{1-474}$ does not maintain a stable hexameric complex in either the wildtype *Pseudomonas* PAO1 strain or in the Δ*fliD*/*fliD$_{1-474}$* complementation strain (*Figure 4e*). The Δ*fliD* transposon strain does not form flagella, as indicated by the lack of FliC in the analyzed

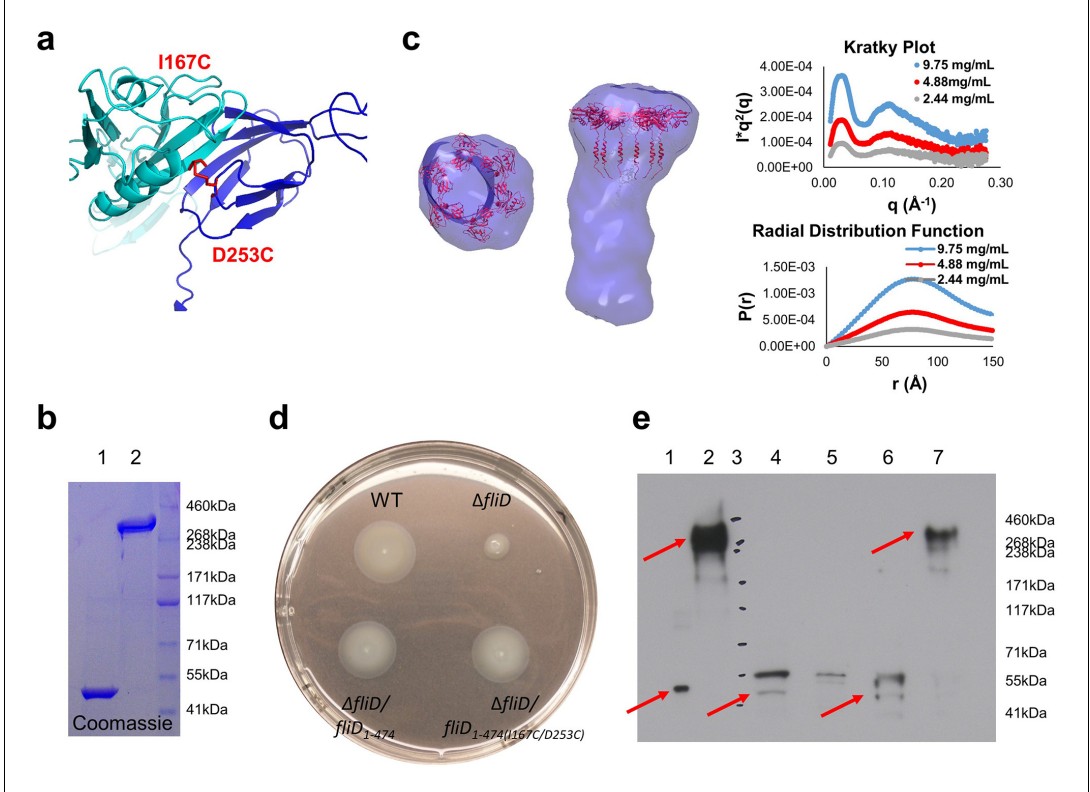

**Figure 4.** Stable hexameric DM1-FliD$_{1-474}$ complements *P. aeruginosa* PAO1 dFliD transposon strain. (**a**) Location of residues I167 and D253, which were predicted by the web server Disulfide by Design 2.0 (*Craig and Dombkowski, 2013*) to form stable disulfide bridges after mutation to cysteines. (**b**) FliD$_{1-474(I167C/D253C)}$ analyzed under reducing (lane 1) and non-reducing (lane 2) conditions by SDS-PAGE. (**c**) SAXS analysis of FliD$_{1-474(I167C/D253C)}$. Kratky plot (*I*q$^2$ *versus q*) and radial distribution function calculated by GNOM for 9.75 mg/mL (blue, used to calculate envelope), 4.88 mg/mL (red) and 2.44 mg/mL (grey). SAXS envelope calculated by DAMMIF with superimposed FliD$_{78-405}$ crystal structure. (**d**) Swimming motility assay of wildtype PAO1 (WT), FliD transposon strain PW2975 (Δ*fliD*), Δ*fliD* complemented with FliD$_{1-474}$ (Δ*fliD/fliD$_{1-474}$*) or FliD$_{1-474(I167C/D253C)}$ (Δ*fliD/fliD$_{1-474(I167C/D253C)}$*), respectively. (**e**) Western blot using anti-FliD scFv-Fc SH1579-B7 showing purified protein FliD$_{1-474(I167C/D253C)}$ under reducing (lane 1) and under non-reducing (lane 2) conditions. The presence of FliD in flagella preparations from wildtype PAO1 (lane 4), Δ*fliD* (lane 5), Δ*fliD/fliD$_{1-474}$*(lane 6) and Δ*fliD/fliD$_{1-474(I167C/D253C)}$* (lane 7) was analyzed under non-reducing conditions. The molecular weight standard is shown in lane 3 and the corresponding molecular weights are indicated on the right side of the blot. The 50 kDa and the 300 kDa bands representing FliD$_{1-474}$ or hexameric FliD$_{1-474(I167C/D253C)}$, respectively, are indicated by red arrows.

The following figure supplements are available for figure 4:

**Figure supplement 1.** Analysis of FliD$_{1-474(I167C/D253C)}$ peptides following pepsin digestion under reducing and non-reducing conditions.

**Figure supplement 2.** Western blot analysis of PAO1 strain flagella preparations.

**Figure supplement 3.** Swimming motility assay.

flagella preparations (lane 5 on the stained Western blot membrane, *Figure 4—figure supplement 2*). In comparison, the wildtype and all complementation strains form flagella as indicated by the presence of flagellin/FliC in the purified flagella samples (*Figure 4—figure supplement 2*). Together, these data indicate that FliD that is covalently locked in its hexameric assembly can form functional flagella that allow *Pseudomonas* bacteria to swim like *Pseudomonas* with wildtype FliD. Thus, the hexamer oligomeric state of *Pseudomonas* FliD is functional in vivo.

We also tested whether *Salmonella* FliD, which is known to form pentamers when capping the flagellar filament, could function as a capping protein for *Pseudomonas* flagella. In contrast to the in vivo functional hexameric forms of *P. aeruginosa* FliD$_{-474}$, complementation of the PAO1 PW2975 transposon strain with *fliD* from *Salmonella typhimurium* (Δ*fliD/fliD$_{StyFliDe}$*) did not restore swimming

motility (*Figure 4—figure supplement 3*). As this clone was codon-optimized for expression in *Escherichia coli*, we also confirmed that a wildtype PAO1 full-length FliD$_{1–474}$ encoded by a similarly codon-optimized gene did restore swimming motility in the ΔfliD strain (*ΔfliD/fliD$_{PAOfliDe}$*; *Figure 4—figure supplement 3*). Although there exist many possible reasons other than oligomeric state that could explain the inability of *Salmonella* FliD to functionally complement *Pseudomonas*, these data suggest that *Pseudomonas* flagella may prefer FliD proteins that adopt hexameric rather than pentameric states.

## Regions outside of the head domains drive FliD oligomerization

Although the FliD$_{78–405}$ crystal structure exhibits intermolecular contacts between the head regions of FliD$_{78–405}$ subunits comprising the hexamer, each of these interfaces is small, with a buried surface area of only 665 Å$^2$, and contains few intermolecular contacts (*Figure 5a*) relative to typical protein–protein interactions (*Jones and Thornton, 1996*). To determine whether these interactions were sufficient to drive oligomerization of FliD, we expressed and purified the head region only, FliD$_{78–273}$. By AUC (*Figure 5b*), chemical crosslinking (*Figure 5c*) and SAXS analysis (*Figure 5d*), we observed that when FliD lacks the leg region and the N- and C-terminal coiled-coil domains, it is present predominantly in the form of monomers (and dimers to a lesser extent) in solution, but fails to form higher-order oligomers as do the longer versions of FliD that we analyzed. Because different buffer

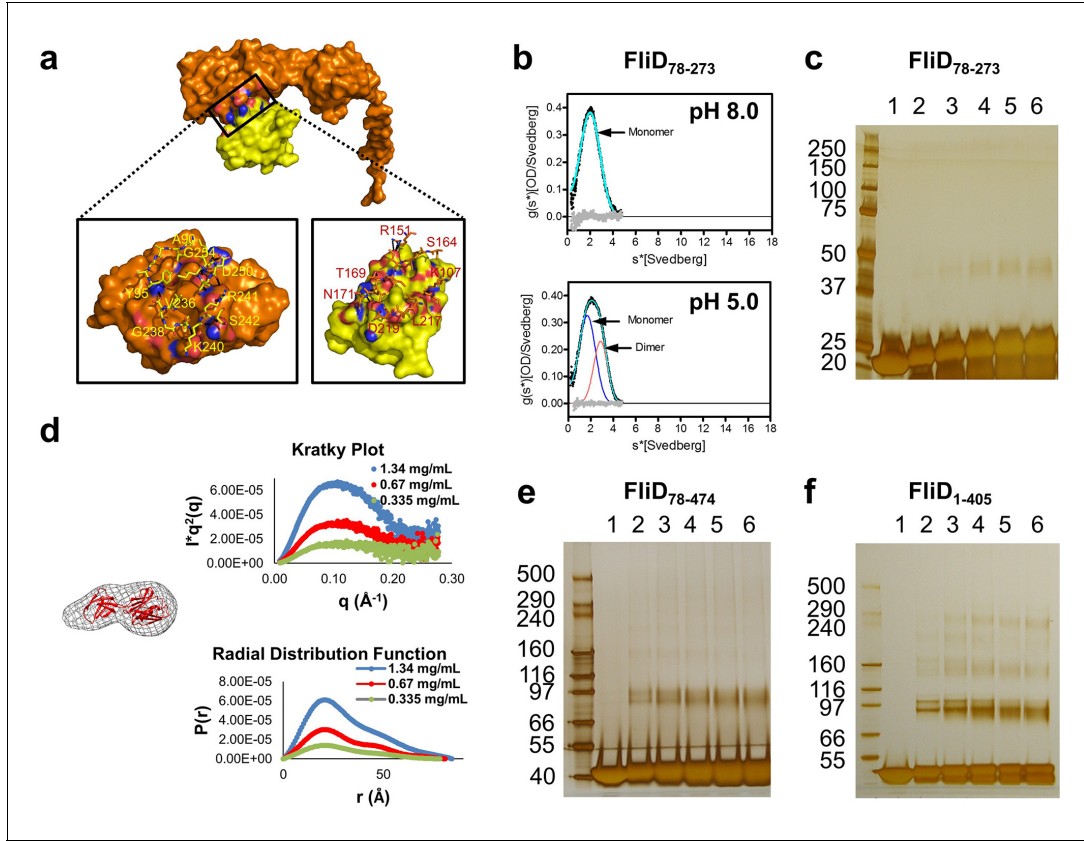

**Figure 5.** Molecular determinants of *Pseudomonas* FliD oligomerization reside outside of the stable head region. (**a**) Intermolecular interface formed between head region monomer subunits, with an 'open book' rendering of the interface expanded below. Head domain 1 is yellow; domain 2 is orange; interface oxygen and nitrogen atoms are red and blue, respectively. (**b**) AUC analysis of the head region alone, FliD$_{78–273}$, at pH 8.0 (upper panel) and pH 5.0 (lower panel) reveals a monomeric species at pH 8.0 and the additional minor presence of a dimeric species at pH 5.0. (**c**) Silver-stained SDS-PAGE after chemical crosslinking of FliD$_{78–273}$. (**d**) SAXS analysis of FliD$_{78–273}$. Kratky plot (*I*q$^2$ *versus q*) and radial distribution function calculated by GNOM and SAXS envelopes calculated by DAMMIF with superimposed crystal structures are shown for FliD$_{7–273}$ at 1.34 mg/mL (blue), 0.67 mg/mL (red, used to calculate the envelope) and 0.335 mg/mL (green). (**e**) Silver-stained SDS-PAGE after chemical crosslinking of FliD$_{78–474}$. (**f**) Silver-stained SDS-PAGE after chemical crosslinking of FliD$_{1–405}$.

conditions, including changes in pH, have been shown to affect the polymerization states of flagellar filaments (*Shibata et al., 2005*) and capping proteins (*Imada et al., 1998*), we performed additional AUC experiments and found that FliD$_{78-273}$ is entirely monomeric at pH 8.0 and becomes approximately one-third dimeric at pH 5.0; we observed no higher-order oligomers of FliD$_{78-273}$ regardless of buffer conditions (*Figure 5b*). We also assessed, by chemical cross-linking, the oligomerization states of FliD variants lacking only the N-terminal coiled-coil domain (FliD$_{78-474}$; *Figure 5e*) or the C-terminal coiled-coil domain (FliD$_{1-405}$; *Figure 5f*). We found them to be mainly monomeric with a minority of species appearing to dimerize, although the latter exhibit weak higher-order oligomerization potential. Kratky plots and radial distribution functions calculated from SAXS data of the variants lacking either the N- or C-terminal coiled-coil domain, FliD$_{78-474}$ or FliD$_{1-405}$ respectively, reveal that these proteins adopt extended shapes with flexible regions that are clearly represented in the resulting molecular envelopes (*Figure 6a,b*). These data indicate that the driving force for hexamerization of *Pseudomonas* FliD resides in molecular determinants outside of the head region and, at a minimum, involves residues in the N-terminal and C-terminal coiled-coil domains. FliD$_{78-405}$ is lacking the C-terminal and N-terminal coiled-coil domain but still assembles into dodecamers, as shown in the crystal structure, cross-linking experiments and AUC, which is likely caused by strand replacement in the head region domain 1 and helix–helix (residues 274–308) interaction of stacked molecules (*Figure 2c*).

## The N- and C-terminal regions of FliD are highly flexible

A large extent of FliD sequence currently remains inaccessible to high-resolution structural analysis, including *Pseudomonas* FliD residues 1–79 and 309–474. Consequently, we performed hydrogen/deuterium (H/D) exchange-mass spectrometry (HDX-MS) experiments with FliD$_{78-405}$ to define its solvent accessible regions and to evaluate its dynamic behavior. We subjected FliD$_{78-405}$ to H/D exchange for 10 s to 2 hr and observed that the head region (residues 80–273) of FliD$_{78-405}$ adopts a largely stable exchange-protected fold with greater stability observed for domain 2 relative to that

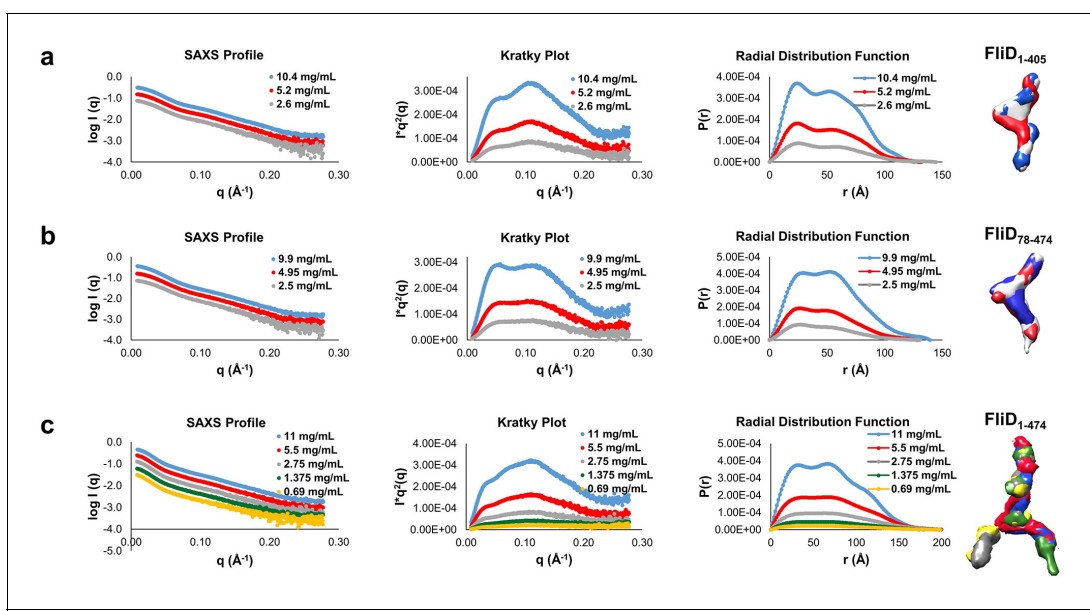

**Figure 6.** Small angle X-ray scattering (SAXS) data of FliD$_{1-405}$, FliD$_{78-474}$ and FliD$_{1-474}$. Log-scale intensity SAXS profiles, Kratky Plot ($I \times q^2$ *versus* $q$), radial distribution function calculated by GNOM and SAXS envelopes calculated by DAMMIF are shown for: (a) FliD$_{1-405}$ at 10.4 mg/mL (blue), 5.2 mg/mL (red) and 2.6 mg/mL (grey); (b) FliD$_{78-474}$ at 9.9 mg/mL (blue), 4.95 mg/mL (red) and 2.5 mg/mL (grey); and (c) FliD$_{1-474}$ at 11 mg/mL (blue), 5.5 mg/mL (red), 2.7 mg/mL (grey), 1.38 mg/mL (green) and 0.69 mg/mL (yellow).
The following figure supplement is available for figure 6:

**Figure supplement 1.** Analytical ultracentrifugation (AUC) analysis of FliD$_{1-474}$ at pH 11.0.

of domain 1. The leg region, particularly residues C-terminal to the α helix observed in the crystal structure, is more disordered or less stable (*Figure 7a*). Residues linking head domain 1 to the leg helix display cooperative unfolding behavior as indicated by EX1 kinetics (*Weis et al., 2006*) that result in double isotopic envelopes (*Figure 7b*). The FliD$_{78-405}$ protein used in these experiments was folded properly as shown by circular dichroism, as were all other FliD protein fragments that we produced (*Figure 7—figure supplement 1*). When mapped to our crystal structure of FliD$_{78-405}$, the degree of H/D exchange over time on the peptide level corresponds to the degree of conformational stability on the residue level (*Figure 7c*). When we used a difference plot to compare the extent of H/D exchange of full length FliD$_{1-474}$ with that of FliD$_{78-405}$, we observed that residues 165–225 exhibit relatively greater stability in FliD$_{78-405}$ (*Figure 8a*). Within this stretch of residues in head domain 2 are those residues, 165–171, which form the interface between the two head region domains in the hexameric complex (*Figure 8b*). We observed an additional region of relative stabilization for residues 298–324, part of which, residues 300–308, correspond to the β strand replacement in the opposing head region domain 1 that drives dodecamer formation of FliD$_{78-405}$ (*Figure 8a*). Kratky plots calculated from SAXS data of full-length FliD$_{1-474}$ at pH 11, which is monomeric under these conditions (*Figure 6—figure supplement 1*), confirms the overall flexible nature of this protein. Accordingly, heterogeneous molecular envelopes calculated from SAXS data collected at different concentrations of FliD$_{1-474}$ also exhibit significant conformational flexibility (*Figure 6c*).

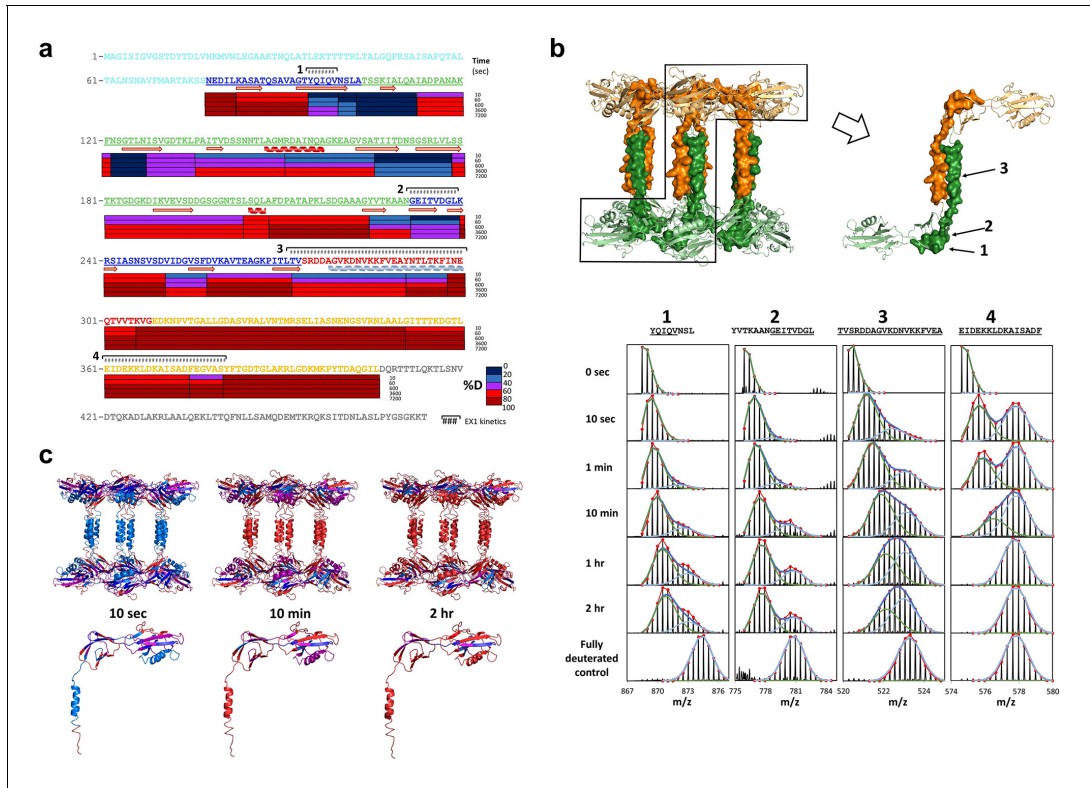

**Figure 7.** Regions of *Pseudomonas* FliD outside of the head domains and initial leg helix are highly dynamic. (**a**) Hydrogen/deuterium exchange analysis of FliD$_{78-405}$. Percent deuteration (%D) heat map is shown. Peptides exhibiting EX1 kinetics are indicated. (**b**) Mass spectra of four FliD peptides exhibit double isotopic envelopes characteristic of EX1 kinetics (*below*). Three of these peptides are mapped to the crystal structure (*above*; FliD hexamers are in green and gold). (**c**) Conformational stability as determined by hydrogen/deuterium exchange mapped to the crystal structure of FliD$_{78-405}$ using the same color coding for %D as shown in (**a**).

The following figure supplement is available for figure 7:

**Figure supplement 1.** Circular dichroism analysis of FliD variants.

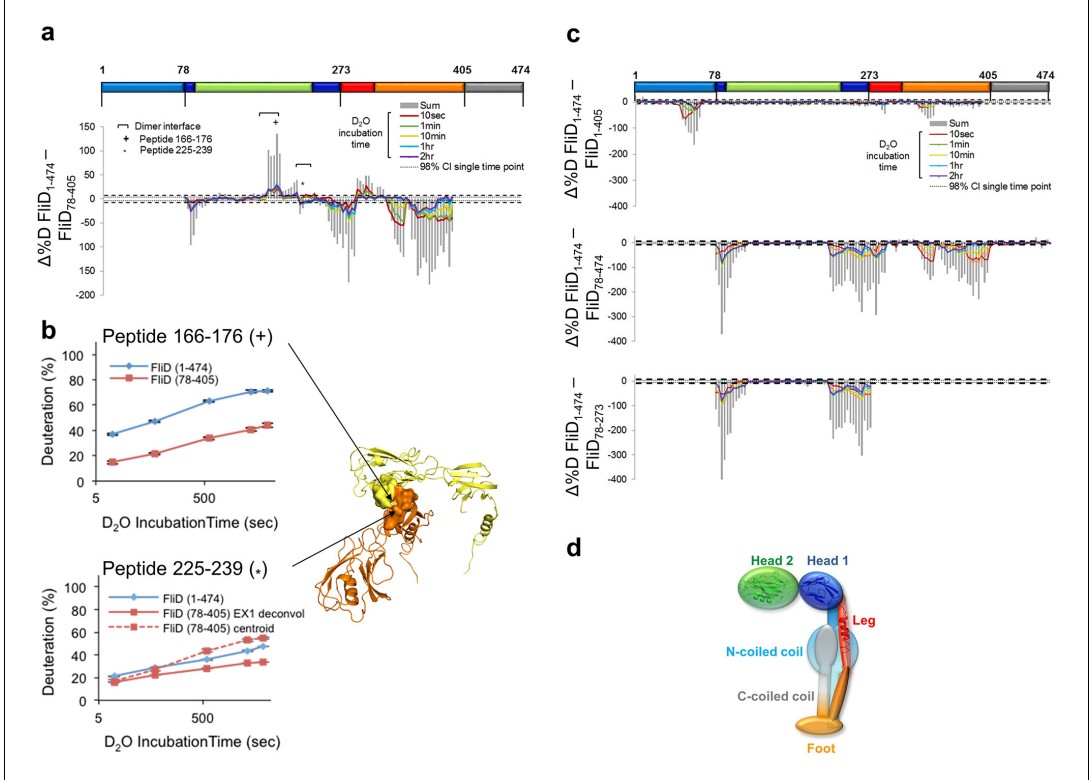

**Figure 8.** Interaction of *Pseudomonas* FliD regions. (**a**) Difference plot of hydrogen/deuterium exchange data from full length $FliD_{1-474}$ and the crystallized fragment, $FliD_{78-405}$. (**b**) Hydrogen/deuterium exchange for peptides corresponding to residues 166–176 (*top*, marked by + in (**a**) and (**b**)) and residues 225–239 (*bottom*, marked by * in (**a**) and (**b**)). Positions of peptides 166–176 and 225–239 in the $FliD_{78-405}$ crystal structure (*right*). (**c**) Difference plots of hydrogen/deuterium exchange data from full length $FliD_{1-474}$ and the fragments missing only the C-terminal coiled coil, $FliD_{1-405}$, only the N-terminal coiled coil, $FliD_{78-474}$, or both the N- and C-terminal coiled coils and the leg domain, $FliD_{78-273}$ (*top, middle and bottom, respectively*). (**d**) Schematic model of the FliD monomeric subunit showing the N-terminal coiled coil stabilizing the head 1 and foot domains and also interacting with the C-terminal coiled coil.

The following figure supplements are available for figure 8:

**Figure supplement 1.** Hydrogen-deuterium exchange-mass spectrometry analysis of FliD variants.

**Figure supplement 2.** Analytical ultracentrifugation (AUC) analysis of $FliD_{1-474}$ at 4 μM.

**Figure supplement 3.** FliD intrinsic disorder analysis.

## Multiple regions of FliD interact with one another

To gain further insight to the dynamics and interactions of the different regions of FliD, we performed HDX-MS experiments with the remaining *Pseudomonas* FliD fragments that we had generated (*Figure 1a*, *Figure 1—figure supplement 1*). HDX-MS heat maps show the overall high degree of flexibility of FliD regions outside the head domain (*Figure 8—figure supplement 1*), and difference plots of FliD truncations in comparison to the full length $FliD_{1-474}$ indicate that the head region, and more specifically head region domain 1, is stabilized by FliD regions outside of the head region (*Figure 8c*). Indeed, the N-terminal coiled coil is responsible for this stabilization of head region domain 1, in addition to stabilization of residues between the leg region α helix and the C-terminal coiled coil (*Figure 8c*). Finally, the C-terminal coiled coil predominantly stabilizes residues in the N-terminal coiled coil (*Figure 8c*). The stabilization of one FliD region by another is most likely to be direct and intramolecular, rather than allosteric and intermolecular, as the full-length protein $FliD_{1-474}$ adopts predominantly monomeric species under the conditions that we used for HDX-MS analysis

(*Figure 8—figure supplement 2*). This leads to a transient structural model of the entire FliD mono-mer subunit (*Figure 8d*) in which the N-terminal coiled coil bridges the head and foot regions and is itself pinned to these structural elements by the C-terminal coiled coil. Notably, none of the flexible regions of FliD, save the C-terminal 20 residues, are consistently predicted by sequence analysis to be intrinsically disordered (*Figure 8—figure supplement 3*); rather, they are inherently capable of adopting a limited number of conformations required to be in an 'on-state' (actively chaperoning and sorting a FliC protein) or an 'off-state' (engaging only structured and previously positioned FliC proteins).

## Discussion

At the gross structural level, our studies show that the oligomeric states differ between FliD protein assemblies in diverse bacteria. *Salmonella* has long served as the model for bacterial flagellum structure and function. Since *Salmonella* FliD performs its native flagellar capping function as a pentamer (*Imada et al., 1998*; *Maki et al., 1998*; *Vonderviszt et al., 1998*; *Yonekura et al., 2000*), it could reasonably be assumed that all FliD proteins form pentamers at the distal ends of all bacterial flagella. We found that *Pseudomonas* FliD instead forms hexamers both in crystals and in solution. Additionally, we showed that *Pseudomonas* FliD constrained to its hexameric state by inter-subunit disulfide bonds is functional in vivo, resulting in the formation of flagella and swimming motility. Conversely, *Salmonella* FliD that assembles as pentamers does not allow flagellar formation and swimming motility in *Pseudomonas* bacteria.

Much like cryo-EM studies of flagellar filaments from diverse bacteria unequivocally showed that the number of protofilaments ranges, at least, from 11 in *Salmonella* (*Yonekura et al., 2003*) to seven in *Campylobacter* (*Galkin et al., 2008*), our structural analyses show that the proteins that cap these filaments also vary in their oligomeric states. The end of the *Salmonella* filament exhibits a non-planar surface with five indentations (*Yonekura et al., 2000*) into which the five legs of the corresponding FliD pentamer have been modeled (*Maki-Yonekura et al., 2003*). Although no structure of the *Pseudomonas* filament has been determined even at low resolution, our hexameric FliD structure suggests that it may incorporate an even greater number of protofilaments than does the *Salmonella* filament. This would allow the formation of an additional molecular cavity on the distal end of FliD that accommodates its unique capping protein hexamer. Indeed, *Pseudomonas* flagellar filaments have been found to be macroscopically different from those of *Salmonella* (*Shibata et al., 2005*).

At 2.2 Å resolution, our crystal structure reveals several previously unknown structural features of FliD that are likely to be critical to its function. First, each one of the six FliD head regions that appear as discreet lobes when visualized by EM analysis is actually composed of the first and second domains of neighboring monomer subunits, as opposed to both domains belonging to the same subunit. This is likely the case for FliD oligomers from diverse bacteria, including *Salmonella* FliD for which cryo-EM analyses showed five head region lobes (*Maki-Yonekura et al., 2003*; *Yonekura et al., 2000, 2003*). Second, despite vanishingly low sequence similarity, each domain within the head region exhibits high structural similarity to the corresponding domains of other flagellar proteins, including the hook-filament junction protein/FlgK and flagellin/FliC. Thus, at least one protein domain from the proteins that occupy the distal ends of the hook (FlgK) and the cap (FliD) adopts a common structural fold, which may be required for and unique to their function at these related positions within the flagellum. This domain conservation among flagellar proteins confirms a previous study proposing a structural relationship of FliD to other flagellar proteins despite their low sequence similarities (*Vonderviszt et al., 1998*). This structural conservation suggests that a structurally similar cap complex may form at both hook and on the distal end. It also suggests that FliD may play a functional role earlier in the flagellar assembly process, prior to its known FliC sorting function. Notably, in mutants of *Salmonella* that lack filaments, FlgK (HAP1), FliD (HAP2) and FlgL (HAP3) form a layered structure at the distal end of the hook-basal body (*Homma and Iino, 1985*). In addition, the shared domain structure of portions of FliC and FliD suggests that the chaperoning activity of FliD could derive, at least in part, from its presentation of a like-structured template against which FliC subunits may fold prior to being positioned into the growing filament.

Owing to the conformational flexibility of FliD necessitated by its FliC sorting function, we still lack high-resolution structural information for certain regions of this protein. However, using a

number of biophysical methods, we found that molecular determinants outside of the conformationally stable head regions control FliD oligomerization. Previous analysis of a trypsinized fragment of *Salmonella* FliD that lacks the N-terminal 42 residues and C-terminal 51 residues, which is similar to the *Pseudomonas* FliD$_{78–405}$ protein that we crystallized, showed that this fragment can form pentamers but not decamers in solution and dissociates into smaller oligomers at low temperatures (*Vonderviszt et al., 1998*). This suggests that determinants outside of the head regions of FliD proteins of diverse bacteria control their oligomeric states. Accordingly, we showed that FliD, when constrained to its hexameric state by disulfide bonds formed between neighboring head domain subunits, is fully functional in vivo. Thus, conformational flexibility in or relative repositioning of the head regions are not functional requirements of FliD but flexibility in regions outside of the head regions undoubtedly is. Considering that inhibiting protein–protein interfaces with small molecules remains a major technical challenge (*Arkin et al., 2014*), the dependence of FliD oligomeric assembly on its flexible regions enhances the prospects of developing small molecule inhibitors of FliD oligomerization, and consequently of flagellar function, as a novel class of antibiotic agents.

## Materials and methods

### Plasmids

For crystallization, the coding sequence optimized for expression in *E. coli* of FliD$_{78–405}$ from the PAO1 strain of *P. aeruginosa* was synthesized and cloned into the pET-28b vector (Novagen) with the inclusion of an N-terminal His$_6$-tag followed by a tobacco etch virus (TEV) protease recognition site. A FliD$_{78–405}$ mutant encoding four leucine to methionine mutations at positions L135, L239, L347 and L350 (FliD$_{78–405}$/L$_4$-M$_4$) was synthesized and likewise cloned into the pET-28b vector with an N-terminal His$_6$-tag. For all other experiments, FliD$_{78–405}$, full-length FliD (FliD$_{1–474}$), head domain-only FliD (FliD$_{78–273}$), and FliD lacking the N-terminal (FliD$_{78–474}$) or C-terminal (FliD$_{1–405}$) coiled-coil domains were codon-optimized for expression in *E. coli* and cloned into pGEX5x2 in frame with an N-terminal GST-tag followed by a TEV protease site. To obtain a stable hexameric full-length FliD mutant (FliD$_{1–474(I167C/D253C)}$), two residues (I167 and D253) located at the interface between neighboring head domains were mutated to cysteine residues in wildtype FliD$_{1–474}$.

### Recombinant protein expression and purification

All FliD constructs from the *P. aeruginosa* PAO1 strain were expressed in LB medium for 4 hr at 37°C in *E. coli* BL21(DE3)pLysS cells after induction with 1 mM IPTG at an OD$_{600nm}$ of 0.6. Selenomethionine (SeMet)-labeled FliD$_{78–405}$/L$_4$-M$_4$ was produced using metabolic inhibition of methionine biosynthesis (*Van Duyne et al., 1993*) and growth in M9 medium containing 60 mg/L SeMet as the sole source of methionine for 6 hr after induction with 1 mM IPTG. Cells were harvested (5000 g for 15 min) and lysed in PBS including 5 mM β-mercaptoethanol by sonication. For crystallization of His-tagged FliD$_{78–405}$ and His-tagged FliD$_{78–405}$/L$_4$-M$_4$, the soluble fraction was purified using HisPur NiNTA Resin (Thermo Scientific). The protein was further purified by size exclusion chromatography (Superdex 200 10/300 GL, GE Healthcare) in PBS followed by anion exchange chromatography (MonoQ 5/50 GL, GE Healthcare). For crystallization the protein was dialyzed into 30 mM Tris pH 8.0, 80 mM sodium chloride and concentrated to approximately 13 mg/mL. FliD-GST-fusion constructs were purified using a Glutathione Sepharose (BioVision) column. Following 16 hr digestion with TEV protease, the GST tag was removed by Glutathione Sepharose and TEV was removed by NiNTA (Thermo Scientific) chromatography. Cleaved FliD constructs were further purified using size exclusion (Superdex 200 10/300 GL, GE Healthcare) in PBS followed by anion exchange chromatography (MonoQ 5/50 GL, GE Healthcare) using 20 mM CHES pH 9.0 and a linear salt gradient from 0 to 1 M NaCl over 12 min.

### Protein crystallization

Crystals of FliD$_{78–405}$ obtained in 0.25 M L-Arginine, 0.1 M Tris/HCl pH 8.0, 8% PGA diffracted poorly and were subsequently used for random microseeding matrix screening (rMMS) (*D'Arcy et al., 2007*). Improved crystals of FliD$_{78–405}$ were grown in 0.8 M NaK Tartrate, 0.1 M Hepes pH 7.5 and diffracted to 2.2 Å. Crystals of SeMet-labeled FliD$_{78–405}$/L$_4$-M$_4$ were also obtained by employing rMMS with the initial, poorly diffracting crystals of FliD$_{78–405}$, which resulted

in $FliD_{78\ -405}/L_4-M_4$ crystals grown in 1.5 M ammonium sulfate, 0.1 M Tris pH 8.5, 10% glycerol diffracting to 3.6 Å (anomalous signal cuttoff). Crystals were harvested and flash cooled in liquid nitrogen in mother liquor supplemented with 25% to 30% glycerol as cryo-protectant.

## X-ray diffraction data processing, structure determination and refinement

X-ray diffraction data for the SeMet-labeled $FliD_{78\ -405}/L_4-M_4$ crystal were collected using a Dectris 6M PILATUS detector on the 12–2 beamline at the Stanford Synchrotron Radiation Lightsource, SSRL, processed using XDS (*Kabsch, 2010b*), scaled in AIMLESS (*Evans and Murshudov, 2013*; *Winn et al., 2011*), and phases obtained using the SSRL multi-wavelength anomalous dispersion (MAD) script by A. Gonzalez with SHELX options based on a script by Qingping Xu, including the programs SHELX (*Schneider and Sheldrick, 2002*), SOLVE (*Terwilliger and Berendzen, 1999*) and RESOLV (*Terwilliger, 2000*). The initial $FliD_{78\ -405}/L_4-M_4$ model was improved manually by rebuilding the peptide chain backbone in Coot (*Emsley and Cowtan, 2004*) and refining using Phenix (*Adams et al., 2010*). Diffraction data for native, wildtype $FliD_{78\ -405}$ were collected using a MARmosaic 300 CCD detector on the 23ID-B beamline at the Advanced Photon Source, Argonne National Laboratory, APS, and processed using XDS (*Kabsch, 2010b*) and XSCALE (*Kabsch, 2010a*). The partially built and refined SeMet-$FliD_{78\ -405}/L_4-M_4$ model was used as a molecular replacement model for phasing the native $FliD_{78-405}$ data using Phaser (*McCoy et al., 2007*). The initial native $FliD_{78-405}$ model was build using Autobuild and improved by manual model rebuilding in Coot (*Emsley and Cowtan, 2004*) and by iterative rounds of refinement using Phenix (*Adams et al., 2010*).

## Mass spectrometry

$FliD_{78-405}$ crystals were crosslinked using 2% formaldehyde, harvested and washed in mother liquor, dissolved in water and the crosslinking reversed by heating the samples to 95°C for 20 min. The samples were analyzed by liquid chromatography (LC)-electrospray ionization (ESI)-mass spectrometry (MS) using a gradient of mobile phase A (0.1% formic acid in water) and mobile phase B (0.1% formic acid in acetonitrile) increasing from 0% B to 90% B in 20 min. The Accela LC System was attached to a LXQ linear ion trap mass spectrometer (Thermo Scientific). Raw MS data were analyzed using Xcalibus Qual Browser (Thermo Scientific) and deconvoluted using BioWorks (Thermo Scientific, Waltham, MA).

## Circular dichroism

10 µM FliD protein in 10 mM sodium phosphate pH 7.0 was used to record a spectrum ranging from 190 nm to 260 nm at 15°C. CD melting curves were analyzed at 222 nm or 205 nm by increasing the temperature by 1°C per minute starting at 15°C using a JASCO J810 CD instrument according to the manufacturer's instructions.

## Electron microscopy

An aliquot of a $FliD_{78-405}$ protein sample was negatively stained with 2% (weight/volume) uranyl acetate and imaged using a Tecnai F20 (FEI) electron microscope operating at 120 keV. Approximately 3500 particles were selected from 70 micrographs and used to generate class averages in EMAN2 (*Tang et al., 2007*). Six classes were generated, and *Figure 3a* shows the single largest class.

## Small angle X-ray scattering (SAXS)

Small angel x-ray scattering data were collected using a dual Pilatus 100K-S SAXS/WAXS detector at beamline G-1 of the Macromolecular Diffraction Facility at the Cornell High Energy Synchrotron Source (MacCHESS). Scattering was measured in 30 mM Tris pH 8.0, 80 mM NaCl of $FliD_{78-405}$ at 10.7 mg/mL, 5.4 mg/mL and 2.7 mg/mL, of $FliD_{78-273}$ at 1.34 mg/mL, 0.67 mg/mL and 0.335 mg/mL, of $FliD_{1-405}$ at 10.4 mg/mL, 5.2 mg/mL and 2.6 mg/mL, of $FliD_{78-474}$ at 9.9 mg/mL, 4.95 mg/mL and 2.6 mg/mL and of $FliD_{1-474(I167C/D253C)}$ at 9.75 mg/ml, 4.88 mg/ml and 2.44 mg/ml. Scattering of monomeric full-length $FliD_{1-474}$ at 11 mg/mL, 5.5 mg/mL, 2.75 mg/mL, 1.38 mg/mL and 0.69 mg/mL was measured in 20 mM CAPS pH 11.0, 80 mM NaCl. The SAXS data were processed using the BioXTAS RAW software (*Nielsen et al., 2009*) and radial distribution functions calculated using GNOM (*Svergun, 1992*). Molecular envelopes were generated using GASBOR (*Svergun et al., 2001*) and

DAMMIF (*Lammie et al., 2007*). FoXS (*Schneidman-Duhovny et al., 2010*) was used to verify the calculated intensity plots of the structures of the head domain $FliD_{78-273}$ and the dodecameric $FliD_{78-405}$. The X-ray crystal structures of $FliD_{78-273}$ and $FliD_{78-405}$ were superimposed onto the envelopes.

## Hydrogen/deuterium exchange-mass spectrometry

The coverage maps for $FliD_{1-474}$ and $FliD_{78-405}$ were obtained from undeuterated controls as follows: 3.5 µL of ~40 µM FliD in 30 mM TrisHCl, 150 mM NaCl pH 8.0 was diluted with 31.5 µL of the same buffer at room temperature followed by the addition of 100 µL of ice cold quench (100 mM Phosphate buffer, 1.5 M Guanidine-HCl, pH 2.4). The quenched samples were injected into a Waters HDX nanoAcquity UPLC (Waters, Milford, MA) with in-line pepsin digestion (Waters Enzymate BEH pepsin column). Peptic fragments were trapped on an Acquity UPLC BEH C18 peptide trap and separated on an Acquity UPLC BEH C18 column. A 7 min, 5% to 35% acetonitrile (0.1% formic acid) gradient was used to elute peptides directly into a Waters Synapt G2 mass spectrometer (Waters, Milford, MA). MS$^E$ data were acquired with a 20 to 30 V ramp trap CE for high energy acquisition of product ions as well as continuous lock mass (Leu-Enk) for mass accuracy correction. Peptides were identified using the ProteinLynx Global Server 2.5.1 (PLGS) from Waters. Further filtering of 0.3 fragments per residues was applied in DynamX.

For each construct, the HD exchange reactions were performed as follows: 3.5 µL of ~40 µM FliD in 30 mM TrisHCl, 150 mM NaCl pH 8.0 was incubated in 31.5 µL of 30 mM TrisDCl, 99.99% $D_2O$, pD 8.0, 150 mM NaCl. All reactions were performed at 25°C. Prior to injection, deuteration reactions were quenched at various times (10 s, 1 min, 10 min, 1 hr and 2 hr) with 100 µL of 100 mM Phosphate buffer, 1.5 Guanidine-HCl, pH 2.4. Back exchange correction was performed against fully deuterated controls acquired by incubating 3.5 µL of 40 µM $FliD_{1-474}$ in 31.5 µL 30 mM TrisDCl, 99.99% $D_2O$, pD 8.0, 150 mM NaCl containing 6 M deuterated Guanidine DCl for 2 hr at 25°C prior to quenching (without guanidine HCl). All deuteration time points and controls were acquired in triplicates.

The deuterium uptake by the identified peptides through increasing deuteration time and for the fully deuterated control was determined using Water's DynamX 2.0 software. The normalized percentage of deuterium uptake (%D) at an incubation time $t$ for a given peptide was calculated as follows:

$$\%D = \frac{100 \cdot (m_t - m_0)}{m_f - m_0}$$

With $m_t$ the centroid mass at incubation time $t$, $m_0$ the centroid mass of the undeuterated control, and $m_f$ the centroid mass of the fully deuterated control. Heat maps and percent deuteration difference plots ($\Delta\%D$) were generated using the percent deuteration calculated. Confidence intervals for the $\Delta\%D$ plots were determined using the method outlined by *Houde et al. (2011)*, adjusted to percent deuteration using the fully deuterated controls. Confidence intervals (98%) were plotted on the $\Delta\%D$ plots as horizontal dashed lines. EX1 type cooperative unfolding was analyzed using HX-Express2 (*Guttman et al., 2013*).

## Determination of peptide coverage of $FliD_{1-474(I167C/D253C)}$ under reducing and non-reducing conditions

Coverage maps of $FliD_{1-474(I167C/D253C)}$ in the presence and absence of reducing agent were obtained similarly as above except for the following: 3 µL of 66 µM $FliD_{1-474(I167C/D253C)}$ were incubated for 2 hr with 15 µL of 8 M Guanidine-HCl and 2 µL of 1 M TCEP (reducing conditions) or 2 µL $H_2O$ (non-reducing conditions). Subsequently, 180 µL of quench buffer (100 mM potassium buffer, pH 2.4) was added and the mixture immediately injected into the Waters HDX nanoAcquity UPLC. The remainder of the workflow, MS method, peptide identification and coverage map determination was unchanged. In addition, Biopharmalynx 1.3.5 (Waters) was used to search for and to identify disulfide-bridged peptides. A filter of 15% b/y ions identified was applied. The search was performed both in the context of the expected C167–C253 disulfide bridge and forthe C167–C167 and C253–C253 disulfide bridges as negative controls.

## Analytical ultracentrifugation

The oligomeric states of $FliD_{78-405}$, $FliD_{78-273}$ and $FliD_{1-474}$ in buffers containing 30 mM Tris, 80 mM NaCl, pH 8.0 or 20 mM sodium citrate, 80 mM NaCl, pH 5.0 respectively, were analyzed by sedimentation velocity using a Beckman-Coulter XL-I analytical ultracentrifuge equipped with a 4- or 8-hole An-60Ti Rotor at 20°C. SedenTerp (http://sednterp.unh.edu) was used to calculate protein partial specific volumes and solvent densities and viscosities from the protein amino acid sequences and buffer compositions. For sedimentation velocity measurements, samples of 295 µM (pH 8.0) or 325 µM (pH 5.0) $FliD_{78-405}$, 124 µM $FliD_{78-273}$, and 168 µM $FliD_{1-474}$ were prepared in each buffer. $FliD_{1-474}$ was also analyzed by AUC at a low concentration of 4 µM in 30 mM Tris, 80 mM NaCl, pH 8.0 and at a concentration of 43 µM at high pH in 20 mM CAPS, 80 mM NaCl, pH 11.0. After exhaustive dialysis to ensure chemical equilibrium, the samples were loaded into cells equipped with 2-hole charcoal-filled epon centerpieces (either 1.2 or 0.3 mm path length) with sapphire windows. Prior to centrifugation, samples were equilibrated in the rotor for at least 2 hr at the desired experimental temperature. Centrifugation was performed at 50,000 ($FliD_{78-273}$), 40,000 ($FliD_{1-474}$) or 30,000 ($FliD_{78-273}$) rpm and scans were acquired at 280 nm. The resulting data were analyzed using DCDT+ version 2.2.1 (*Philo, 2006*; *Stafford, 1997*, *1992*) to determine the number of species, their sedimentation coefficients, and their fractional contributions to the species populations. All sedimentation coefficients were corrected to $s_{20,w}$ values.

## Chemical crosslinking

Approximately 0.4 mg/mL of protein in 20 mM Hepes pH 8.0, 10 mM sodium chloride was crosslinked using 20 mM 1-ethyl-3-(3-dimethylaminopropyl)carbodiimide hydrochloride (EDC) and 20 mM N-hydroxysuccinimide (NHS) in 20 mM sodium phosphate, pH 7.0, 150 mM NaCl for various time points. The reaction was stopped by the addition of 0.5 M Tris-HCl pH 8.0 to a final concentration of 0.25 M. The products were analyzed on NuPAGE 3–8% Tris-Acetate gels (Life Technologies, Carlsabd, CA) or an Any kD Mini-PROTEAN TGX gels (BioRad) using a silver-staining kit (Thermo Scientific).

## Complementation of *Pseudomonas aeruginosa* PAO1

Wildtype $fliD_{1474}$ and $fliD_{1474(I167C/D253C)}$ were cloned into pUCP20 and transformed by electroporation (*Cadoret et al., 2014*) into the $\Delta fliD$ transposon strain PW2975 (obtained from the Manoil Lab at the University of Washington), resulting in the strains $\Delta fliD/fliD_{1-474}$ and $\Delta fliD/fliD_{1-474(I167C/D253C)}$, respectively. Wildtype $fliD_{1-474}$ ($fliD_{1-474e}$) and full-length $fliD$ from *Salmonella typhimurium* ($fliD_{StyFliDe}$) both with genes codon-optimized for *E. coli* expression were also transformed into PW2975 resulting in $\Delta fliD/fliD_{1-474e}$ and $\Delta fliD/fliD_{StyFliDe}$, respectively.

## Swimming motility assays

Swimming motility assays of *Pseudomonas aeruginosa* strains were performed as described by *Ha et al. (2014)*.

## Isolation of flagella and FliD detection

*Pseudomonas aeruginosa* PAO1 was grown overnight in LB liquid culture, cells were spun down and resuspended in PBS. Flagella were sheared off the cells by passing the suspension through a 23 gauge needle approximately 25 times. After centrifugation, the supernatant containing flagella was concentrated, proteins separated by SDS-PAGE and analyzed by Western blot using anti-FliD scFv-Fc SH1579-B7 and an anti-human-IgG-HRP conjugate secondary antibody.

## Generation of anti-FliD antibodies

Human antibodies were generated as described by *Frenzel et al. (2014)*. In brief, recombinant head region only $FliD_{78-273}$ was immobilized on Costar High Binding plates and incubated with the Hyperphage packaged human antibody gene libraries HAL9/10 (*Kügler et al., 2015*) for negative selection. The non-binding scFv phages were incubated with recombinant full-length $FliD_{1-474}$ to select binders specific for the leg region of FliD. In total, three panning rounds were performed and monoclonal antibodies were identified as described by *Frenzel et al. (2014)*. The antibody SH1579-B7

was recloned as an scFv-Fc (Yumab) with a human IgG1 Fc region and produced in mammalian cell culture as described by *Jäger et al. (2013)*.

## Intrinsic disorder

The sequence of the full-length FliD was submitted to twelve publicly available servers implementing different algorithms for protein disorder prediction. In all cases, we used the default parameters. The servers used were as follows: disEMBL (*Linding et al., 2003a*), GlobProt (*Linding et al., 2003b*), IUPred (*Dosztanyi et al., 2005*), RONN (*Yang et al., 2005*), DisPro (*Cheng et al., 2005*), PONDR (*Romero et al., 2001*), Spine-D (*Zhang et al., 2012*), OnD-CRF (*Wang and Sauer, 2008*), Foldindex (*Prilusky et al., 2005*), MFDp (*Mizianty et al., 2010*), MFDp2 (*Mizianty et al., 2010*), and MD2 (*Kozlowski and Bujnicki, 2012*). Averaging of the results gave all servers equal weight.

## Accession code

Coordinates and structure factors have been deposited in the Protein Data Bank under accession code 5FHY.

## Acknowledgements

We thank the staff of the Stanford Synchrotron Radiation Lightsource (SSRL) beamline 12-2, SLAC National Accelerator Laboratory, San Francisco, USA, of the Advanced Photon Source (APS) GM/CA CAT beamline 23-ID-B, Argonne National Laboratory, Illinois, USA and of the beamline G-1 of the Macromolecular Diffraction facility at Cornell High Energy Synchrotron Source (MacCHESS), Ithaca, USA for their support. This work is supported in part by the University of Maryland Baltimore, School of Pharmacy Mass Spectrometry Center (SOP1841-IQB2014).

## Additional information

### Competing interests

EHE: Reviewing editor, *eLife*. The other authors declare that no competing interests exist.

### Funding

| Funder | Grant reference number | Author |
|---|---|---|
| National Center for Research Resources | NIH S10 RR15899 | Dorothy Beckett |

The funders had no role in study design, data collection and interpretation, or the decision to submit the work for publication.

### Author contributions

SP, Designed the project, Performed experiments, Analysis of data, Writing the manuscript, Conception and design, Acquisition of data, Analysis and interpretation of data, Drafting or revising the article; DD, Performed experiments, Analysis of data, Reviewed the manuscript, Acquisition of data, Analysis and interpretation of data; DAB, KD, PLW, Analysis of data, Analysis and interpretation of data; XY, MH, DB, Performed experiments, Analysis of data, Acquisition of data, Analysis and interpretation of data; SH, Performing experiments, Acquisition of data; AV, AF, Intrinsic disorder prediction, Analysis and interpretation of data; EHE, Performed experiments, Analysis of data, Analysis and interpretation of data; EJS, Designed the project, Analysis of data, Writing the manuscript, Conception and design, Analysis and interpretation of data, Drafting or revising the article

### Author ORCIDs

Sandra Postel, http://orcid.org/0000-0002-6717-1870
Edward H Egelman, http://orcid.org/0000-0003-4844-5212
Eric J Sundberg, http://orcid.org/0000-0003-0478-3033

## Additional files

### Major datasets

The following dataset was generated:

| Author(s) | Year | Dataset title | Dataset URL | Database, license, and accessibility information |
|---|---|---|---|---|
| Postel S, Bonsor D, Diederichs K, Sundberg EJ | 2016 | Crystal structure of FliD (HAP2) from Pseudomonas aeruginosa PAO1 | http://www.rcsb.org/pdb/explore/explore.do?structureId=5FHY | Publicly available at the RCSB Protein Data Bank (accession no. 5FHY) |

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
