## [Decision Letter]

Thank you for submitting your article "Bacterial flagellar capping proteins adopt diverse oligomeric states" for consideration by *eLife*. Your article has been favorably evaluated by Richard Losick (Senior Editor) and three reviewers, one of whom, Richard M Berry (Reviewer #1), is a member of our Board of Reviewing Editors. The following individuals involved in review of your submission have agreed to reveal their identity: Keiichi Namba (Reviewer #2) and Gillian Fraser (Reviewer #3).

The reviewers have discussed the reviews with one another and the Reviewing Editor has drafted this decision to help you prepare a revised submission.

All reviewers agree that the findings are important and suitable for publication in *eLife*. However, one major point of concern remains: whether the hexameric form is biologically relevant in the full-length FliD, rather than a consequence of the loss of the D0 domain.

Before the paper can be accepted for publication, the authors should present solid experimental evidence that the hexameric complex is really the physiological form of the filament cap of *Pseudomonas* flagella. This might best be achieved if the authors could show hexamers of full-length FliD by negative stain EM and class averaging, as they have done for the FliD_78-405_ truncate.

The individual reviewers' reports are included below. The authors should also address the reviewers' points in detail.

Reviewer #1:

The presentation of the atomic structure of the hexameric fragment of FliD, with its similarities in fold to other flagellar components but different symmetry to the pentameric FliD of *Salmonella* make this paper interesting and important.

Minor comments:

1) Introduction: Chain mechanism is controversial. Acknowledge that and cite instead – Trends Microbiol. 2015 May;23(5):296-300. Fueling type III secretion. Lee PC, Rietsch A. Or omit entirely (see reviewer #2, comment 2).

2) In the last paragraph of the subsection “FliD from *P. aeruginosa* PAO1 forms a hexamer”: Where does the finding of a Tetramer by analytic ultracentrifugation in the full-length protein fit in the model? Present evidence of a hexamer in the full-length protein.

3) In the subsection “Multiple regions of FliD interact with one another”: show a model of a full-length monomer and relate this to the native state.

4) In the first paragraph of the Discussion: Is this the most likely solution? The *Salmonella* filament has 5, 6 and 11-start helices, so could just as easily accommodate 6 as 5? (See also reviewer #3, comment 1).

5) In the first paragraph of the Discussion: Compare to "domain-swap" model of FliG – Domain-swap polymerization drives the self-assembly of the bacterial flagellar motor. Baker MA et al. Nat Struct Mol Biol. 2016 Mar;23(3):197-203. But given that oligomerization interactions are thought to reside elsewhere, are these also domain-swapped? Any evidence to this, or too much missing structure?

Reviewer #2:

This paper describes the structural study of *Pseudomonas* FliD by X-ray crystallography and biophysical analyses by multiple methods. FliD is a protein that forms the cap complex at the distal end of the bacterial flagellar filament for the promotion of filament assembly by many copies of flagellin exported from the cytoplasm. The filament does not form in the absence of the FliD cap. This study presents the first high-resolution structure of FliD from any bacterial species. It also describes the hexameric complex structure, which is in contrast with the pentameric complex formed by *Salmonella* FliD, and the intermolecular interactions involved in the hexameric state. The authors discuss the implication of the difference in the stoichiometry between bacterial species.

Minor comments:

1) Abstract: Phrases such as "regulates assembly" and "Without FliD, flagella are improperly constructed" do not appropriately describe the FliD function because the flagella are not formed at all in the absence of FliD.

2) Introduction: Many people have expressed doubt on the validity of the chain mechanism proposed by Evans et al. 2013, partly because of the presence of solid data of an exponential decay in the flagellar growth rate by Iino 1974. In any case the chain mechanism is irrelevant to this study.

3) In the subsection “Crystal structure of the FliD protein from *P. aeruginosa* PAO1” the third sentence from the end is a bit difficult to follow.

4) In the second paragraph of the Discussion: Although FlgK binds directly to the distal end of the hook, it is not the hook capping protein but is a junction protein that connects the hook and filament.

5) In the subsection “Analytical Ultracentrifugation”: Use of a program Sedfit is recommended for the data analysis of analytical ultracentrifugation.

6) Figure 1–Figure 3: Domain labels of Head 1 and Head 2 for model figures would help readers to identify them.

7) Figure 3–Figure 5: The units of q and r are all missing.

8) Figure 6 description of the color code is missing.

9) Figure 7: Since there are five panels in (A), it is difficult to identify which are "above", "below", and "right". There is no figure (C).

Reviewer #3:

Sandra Postel and colleagues present the atomic structure of a major central fragment of *Pseudomonas aeruginosa* FliD, the capping protein for the flagellar filament. This first high-resolution structure of FliD reveals that the cap monomer, which oligomerises to form an annular structure, shows unexpected structural similarity to the axial components of the flagellum, which polymerise to form hollow helical structures. These novel findings are the major strengths of the manuscript.

The FliD_78-405_ truncate packs as a hexamers and dodecamers in the crystal lattice, and the authors go on to show that these hexameric and dodecameric forms are also found in solution, as shown by negative stain EM class averaging. This finding is supported by data from AUC and crosslinking (the SAXS data support formation of a hollow sphere that is not at odds with the dodecameric form). The physiologically relevant hexameric form of *Pseudomonas* FliD is different from the *Salmonella* FliD pentamer observed by Namba and colleagues using a range of methods, including cryo-EM.

1) My main concern is that the hexameric form of *Pseudomonas* FliD might be an artifact caused by truncation of the termini (residues 1-77 and 406-474), removing the flexible D0 domains so critical to FliD oligomerization. The authors do perform AUC and crosslinking with full-length *Pseudomonas* FliD that demonstrate formation of higher order oligomers, but I'm not convinced that these data support the proposal that the hexamer or dodecamer are the predominant forms. I would be more convinced if the authors could show hexamers of full-length FliD by negative stain EM and class averaging, as they have done for the FliD_78-405_ truncate.

A couple of published findings about the *Pseudomonas* flagellar cap and filament further increase my doubts about the physiological relevance of the FliD hexamer. The first is the finding by Aizawa and colleagues that *Pseudomonas* fliD, when expressed in trans, can complement the filament assembly defect of the *Salmonella* fliD null strain (Exchangeability of the flagellin (FliC) and the cap protein (FliD) among different species in flagellar assembly. Inaba S, Hashimoto M, Jyot J, Aizawa S. (2013) Biopolymers. 99(1):63-72. doi: 10.1002/bip.22141). The second is the finding by Trachtenberg and colleagues that the flagellar filament of *Pseudomonas r*hodos appears to have 11-fold symmetry, similar to the *Salmonella* filament (The axial α-helices and radial spokes in the core of the cryo-negatively stained complex flagellar filament of *Pseudomonas* rhodos: recovering high-resolution details from a flexible helical assembly. Cohen-Krausz S, Trachtenberg S. (2003) J Mol Biol 331(5):1093-108).

In my view, stronger data on the oligomeric form of full-length *Pseudomonas* FliD is required.

[Editors' note: further revisions were requested prior to acceptance, as described below.]

Thank you for resubmitting your work entitled "Bacterial flagellar capping proteins adopt diverse oligomeric states" for further consideration at *eLife*. Your revised article has been favorably evaluated by Richard Losick (Senior Editor) and a Reviewing Editor.

The manuscript has been improved but there are some remaining issues that need to be addressed before acceptance, as outlined below:

1) The experiment demonstrating that *Salmonella* FliD fails to complement *Pseudomonas* is a suitable response to a comment of reviewer #3. Without mention of this comment it still warrants inclusion in the text, but the statement of the motivation for the experiment (subsection “FliD from *P. aeruginosa* PAO1 forms a hexamer”, last paragraph) is too strong. As written, this calls into doubt the new result of Figure 4, which answers the reviewers' main objection to the original manuscript. Please either explain how a pentameric *Pseudomonas* FliD might be compatible with Figure 4, or remove this statement.

2) Please acknowledge that there are many possible reasons other than oligomeric state that might explain the inability of *Salmonella* FliD to complement *Pseudomonas*.

---

## [Author Response]

All reviewers agree that the findings are important and suitable for publication in eLife. However, one major point of concern remains: whether the hexameric form is biologically relevant in the full-length FliD, rather than a consequence of the loss of the D0 domain.

Before the paper can be accepted for publication, the authors should present solid experimental evidence that the hexameric complex is really the physiological form of the filament cap of Pseudomonas flagella. This might best be achieved if the authors could show hexamers of full-length FliD by negative stain EM and class averaging, as they have done for the FliD_78-405_ truncate.

Reviewer #1:

Minor comments:

*1) Introduction: Chain mechanism is controversial. Acknowledge that and cite instead – Trends Microbiol. 2015 May;23(5):296-300. Fueling type III secretion. Lee PC, Rietsch A. Or omit entirely (see reviewer #2, comment 2).*

We have removed the sentence that refers to the chain mechanism.

*2) In the last paragraph of the subsection “FliD from P. aeruginosa PAO1 forms a hexamer”: Where does the finding of a Tetramer by analytic ultracentrifugation in the full-length protein fit in the model? Present evidence of a hexamer in the full-length protein.*

In the absence of the flagellar filament, full-length FliD from *Pseudomonas* PAO1 does not predominate as stable hexamers. We have shown that the hexamer is formed in crosslinking experiments and by AUC, but only a small fraction of the total protein actually forms a hexamer. By crosslinking, we can detect monomeric, dimeric, trimeric, tetrameric, pentameric and hexameric species; by AUC analysis of highly concentrated FliD solutions, we detect tetrameric and hexameric species, predominantly. It is possible that the tetramer may be some form of stable intermediate on the path to hexamers, but we do not have enough data to say this unequivocally. Despite the lack of stability of the *Pseudomonas* FliD hexamer, we previously (prior to submission of the original manuscript, in fact) attempted to perform negative-stain EM using both crosslinked FliD_1-474_ and native FliD_1-474_ at low pH, in the hopes that we could detect larger particles. Unfortunately, we were unsuccessful in this regard. The tendency of *Pseudomonas* FliD to oligomerize is much less than that of *Salmonella* FliD, for which the equilibrium state in solution, even in the absence of the flagellar filament, is almost entirely decameric, as previously shown by crosslinking and AUC analyses conducted by others (and confirmed by us – data not shown).

Regardless of whether we could visualize the *Pseudomonas* FliD hexamer or not in solution at low resolution by negative-stain EM, we contend that this experiment would not conclusively show that the hexamer was the bona fide functional unit. To show this, we would have to show that *Pseudomonas* FliD acts as a hexamer in vivo, which we have done in a series of additional experiments shown in a new Figure 4. These experiments are briefly summarized, as follows. We designed, based on our crystal structure, disulfide bonds in full-length *Pseudomonas* FliD that results in a stable FliD hexamer under non-reducing conditions, which we showed by SDS-PAGE, mass spectrometry and SAXS. We then complemented a △*fliD Pseudomonas* strain with the full-length disulfide-constrained *fliD_1-474(I167C/D253C)_* gene, which we tested for swimming motility in a functional assay. This complemented strain restored the swimming motility that was lost in the △*fliD* strain, in a way indistinguishable to that of a full-length wildtype complement (△*fliD/fliD_1-474_*). We isolated flagella from the live, growing △*fliD/fliD_1-474(I167C/D253C)_ Pseudomonas* bacteria and showed by Western blot analysis that the FliD_1-474(I167C/D253C)_ protein remained hexameric in these flagella that provided swimming motility to the bacteria. We therefore conclude that FliD from *Pseudomonas* is functional in a disulfide-locked, hexameric state. Because the disulfide bond was formed between neighboring head regions, these data also indicate that flexibility of the head domains, or at least their relative repositioning, is not required for FliD capping function. In contrast, we found in a similar set of genetic and functional experiments that *Salmonella* FliD cannot complement a △*fliD Pseudomonas* strain and does not restore swimming motility, additional evidence that FliD in a pentameric state is incompatible with *Pseudomonas* flagella formation.

*3) In the subsection “Multiple regions of FliD interact with one another”: show a model of a full-length monomer and relate this to the native state.*

We have added a model of what our data suggests the full-length monomer structure is (see Figure 8). This model is based on the following data: (i) our crystal structure indicating the atomic positions of FliD residues 78-273 (Figure 1); (ii) the likely interfacing regions from those domains that are not modeled in our crystallographic model or missing from the fragment that we crystallized, including the foot, N-terminal coiled coil and C-terminal coiled coil domains, as indicated by our HDX-MS data (Figure 8); and (iii) the SAXS molecular envelope that we generated for the disulfide-linked hexamer (Figure 4). We believe that this model of the monomer reflects its native “resting” state, that is, the structure of the monomer in the absence of the flagellar filament or of a FliD monomer subunit on the distal end of the flagellar filament that is not actively chaperoning and sorting a FliC protein subunit. The dynamic nature of FliD suggests that FliD monomers within the cap oligomer will adopt a different structure(s) when chaperoning and sorting FliC subunits, which our data does not inform and that we believe is beyond the scope of this paper.

*4) In the first paragraph of the Discussion: Is this the most likely solution? The Salmonella filament has 5, 6 and 11-start helices, so could just as easily accommodate 6 as 5? (See also reviewer #3, comment 1).*

Yes, all of our data indicates that *Pseudomonas* FliD oligomerizes as a hexamer in crystals, in solution and in vivo (see response to reviewer #1, comment #2). No evidence exists that *Salmonella* FliD ever adopts any form other than a pentamer (except when it forms a decamer in the absence of the flagellar filament) and, likewise, there is no evidence that *Pseudomonas* FliD exists in a state other than a hexamer in vivo.

*5) In the first paragraph of the Discussion: Compare to "domain-swap" model of FliG – Domain-swap polymerization drives the self-assembly of the bacterial flagellar motor. Baker MA et al. Nat Struct Mol Biol. 2016 Mar;23(3):197-203. But given that oligomerization interactions are thought to reside elsewhere, are these also domain-swapped? Any evidence to this, or too much missing structure?*

We have shown that *Pseudomonas* FliD oligomerization is not driven by interactions between the head domains. Accordingly, “domain-swapping” between head domains cannot be the driving force for self-assembly. Also, by locking full-length FliD in a hexameric state with disulfide bonds between neighboring head domains, the protein remains functional and complements △*fliD* strains in swimming motility assays. Therefore, it is also unlikely that domain-swap polymerization between the head domains plays any functional role. Without high-resolution structural information for the other flexible regions of FliD that do drive self-assembly, though, it is possible that domain-swaps occur between these other domains. Such a domain-swap molecular mechanism within these other regions outside of the head region may contribute, in part or in whole, to FliD self-assembly. However, with such an extent of missing structure, we have no evidence of this at this time.

Reviewer #2:

*Minor comments:*

*1) Abstract: Phrases such as "regulates assembly" and "Without FliD, flagella are improperly constructed" do not appropriately describe the FliD function because the flagella are not formed at all in the absence of FliD.*

This has been altered accordingly in text.

*2) Introduction: Many people have expressed doubt on the validity of the chain mechanism proposed by Evans et al. 2013, partly because of the presence of solid data of an exponential decay in the flagellar growth rate by Iino 1974. In any case the chain mechanism is irrelevant to this study.*

This sentence has been removed (see comment to reviewer #1, comment #1).

*3) In the subsection “Crystal structure of the FliD protein from P. aeruginosa PAO1” the third sentence from the end is a bit difficult to follow.*

We have split this long, difficult to follow sentence into two sentences, which now reads:

“Both analyses indicated that the crystals consisted of an approximate 50:50 mixture of the FliD_78-405_ protein used for crystallization and a further proteolyzed version with a molecular weight of about 27 kDa. The N-terminal His_6_-tag is still detectable by Western blot indicating that proteolysis occurred from the C-terminus (Figure 1—figure supplement 2).”

*4) In the second paragraph of the Discussion: Although FlgK binds directly to the distal end of the hook, it is not the hook capping protein but is a junction protein that connects the hook and filament.*

This has been updated in text.

*5) In the subsection “Analytical Ultracentrifugation”: Use of a program Sedfit is recommended for the data analysis of analytical ultracentrifugation.*

We used DC/DT+, which generally performs as well as SedFit in analysis of sedimentation velocity data, to analyze our data. We used SedFit to analyze some of the data and obtained results essentially identical to those we obtained using DC/CT+. Since we found no difference in the analyses, we have not changed the description of how we handled the data in the text.

*6) Figure 1–Figure 3: Domain labels of Head 1 and Head 2 for model figures would help readers to identify them.*

Domain labels have been added to Figure 1 and 2.

*7) Figure 3–Figure 5: The units of q and r are all missing.*

The units of q and r have now been added to these figures.

*8) Figure 6 description of the color code is missing.*

A description of the color code has been added to Figure 7 (formerly Figure 6). The color code in Figure 7 is the same as in Figure 7, which has been added to legend.

*9) Figure 7: Since there are five panels in (A), it is difficult to identify which are "above", "below", and "right". There is no figure (C).*

Figure 8 (formerly Figure 7) has been added. We also added a new Figure 8 in an attempt to avoid some of the confusion caused by our previous labeling scheme.

Reviewer #3:

*1) My main concern is that the hexameric form of Pseudomonas FliD might be an artifact caused by truncation of the termini (residues 1-77 and 406-474), removing the flexible D0 domains so critical to FliD oligomerization. The authors do perform AUC and crosslinking with full-length Pseudomonas FliD that demonstrate formation of higher order oligomers, but I'm not convinced that these data support the proposal that the hexamer or dodecamer are the predominant forms. I would be more convinced if the authors could show hexamers of full-length FliD by negative stain EM and class averaging, as they have done for the FliD_78-405_ truncate.*

A couple of published findings about the Pseudomonas flagellar cap and filament further increase my doubts about the physiological relevance of the FliD hexamer. The first is the finding by Aizawa and colleagues that Pseudomonas fliD, when expressed in trans, can complement the filament assembly defect of the Salmonella fliD null strain (Exchangeability of the flagellin (FliC) and the cap protein (FliD) among different species in flagellar assembly. Inaba S, Hashimoto M, Jyot J, Aizawa S. (2013) Biopolymers. 99(1):63-72. doi: 10.1002/bip.22141). The second is the finding by Trachtenberg and colleagues that the flagellar filament of Pseudomonas rhodos appears to have 11-fold symmetry, similar to the Salmonella filament (The axial α-helices and radial spokes in the core of the cryo-negatively stained complex flagellar filament of Pseudomonas rhodos: recovering high-resolution details from a flexible helical assembly. Cohen-Krausz S, Trachtenberg S. (2003) J Mol Biol 331(5):1093-108).

Full-length *Pseudomonas* FliD is not stable enough to perform negative stain EM and class averaging as we did with the FliD_78-405_ fragment for which we also solved the crystal structure. Instead, we have shown that the hexameric form of *Pseudomonas* FliD is functional in vivo, a more rigorous and realistic test of whether the hexameric state is not simply an artifact caused by truncation or other manipulation of the protein.

We have carefully reviewed the Inaba et al. paper to which the reviewer refers. We conclude that the authors of this paper have incorrectly interpreted their data. While it is clear from their data that *Salmonella* and *E. coli* FliC and FliD can be exchanged (see Figures 2A, 3A, 4A and 7A), it is equally clear that *Salmonella* and *Pseudomonas* FliC as well as FliD cannot be exchanged to produce functional flagella. For instance, there is clearly no difference in Figure 2B, spots #3 (*Salty* △*fliC* mutant (SJW2536)), #4 (*Salty* △*fliC* mutant transduced with p*Pseae* (PAO1) *fliC*). #5 (*Salty* △*fliC* mutant p*Pseae* (PAK) *fliC*), #6 (*Salty* △*fliD* mutant (MH43)), and#7 (*Salty* △*fliD* mutant transduced with p*Pseae* (PAO1) *fliD*). None of these strains exhibit swarming motility in this assay, whereas wildtype *Pseae* (PAK) and *Pseae* (PAO1) strains do exhibit swarming motility, as seen in Figure 2B, spots #1 and #2, respectively. Likewise, the electron microscopic evidence is non-existent for functional FliC and FliD exchange between *Salmonella* and *Pseudomonas*. In Figure 5A, panels #2 (*Salty* △*fliC* mutant transduced with p*Pseae* (PAO1) *fliC*) and #3 (*Salty* △*fliC* mutant transduced with p*Pseae* (PAK) *fliC*) do not show functional flagella. Similarly, appendages shown in Figure 8A (*Salty* △*fliD* mutant transduced with p*Pseae* (PAO1) *fliD*) look nothing like fully formed wildtype flagella. These short, stubby appendages do not resemble the functional flagella shown in other electron micrographs in this paper (e.g., of wildtype bacteria strains), but could possibly be flagella formed up to the hook, which would fit with non-exchangeability of these FliC and also FliD proteins (although we do not have access to enough of their data to say this with any certainty). Finally, in Figure 9C, the double complementation (*Salty* △*fliC* △*fliD* mutant transduced with both p*Pseae* (PAO1) *fliC* and p*Pseae* (PAO1) *fliD* together) exhibit absolutely no appendages that could be (mis)construed as flagella. It is unclear to us how Inaba et al. could interpret these data to conclude that *Salmonella* and *Pseudomonas* FliD are functionally exchangeable.

We took the additional step of testing this concept ourselves. We asked whether *Pseudomonas* FliD could be functionally exchanged with *Salmonella* FliD, using wildtype *Pseudomonas*, a △*fliD* mutant *Pseudomonas* strain, and the same △*fliD* mutant *Pseudomonas* strain complemented with *Salmonella fliD* – essentially a repetition of the Inaba et al. experiments but in the reverse knock-out/complementation direction. In our swimming motility assay, we found that the latter (△*fliD/fliD_StyFliDe_*) strain was completely non-functional in this assay (see Figure 4—figure supplement 3 in our revised manuscript). These data agree with our re-interpretation of Inaba et al.’s data and indicate that, indeed, *Salmonella* FliD (a pentamer) cannot be functionally exchanged with *Pseudomonas* FliD (a hexamer).

We have also revisited the Cohen-Krausz & Trachtenberg paper mentioned by the reviewer. Well prior to the publication of this paper in 2003, *Pseudomonas rhodos* had been renamed *Methylobacterium rhodinum*. These bacteria diverge at the class level – the former belongs to the class of Gammaproteobacteria and the latter to Alphaproteobacteria. No genomic sequence data is available for this bacterium and no gene or protein sequence from it appears when we conduct a BLAST search using the *Pseudomonas fliD* gene or FliD protein sequences. Thus, we do not believe that this bacterium and whatever morphology their flagella may or may not adopt is relevant to our studies.

[Editors' note: further revisions were requested prior to acceptance, as described below.]

The manuscript has been improved but there are some remaining issues that need to be addressed before acceptance, as outlined below:

1) The experiment demonstrating that Salmonella FliD fails to complement Pseudomonas is a suitable response to a comment of reviewer #3. Without mention of this comment it still warrants inclusion in the text, but the statement of the motivation for the experiment (subsection “FliD from P. aeruginosa PAO1 forms a hexamer”, last paragraph) is too strong. As written, this calls into doubt the new result of Figure 4, which answers the reviewers' main objection to the original manuscript. Please either explain how a pentameric Pseudomonas FliD might be compatible with Figure 4, or remove this statement.

We have removed the statement of the motivation for the experiment, which we agree was too strong and confusing. The paragraph now starts “We also test whether *Salmonella* FliD…”.

2) Please acknowledge that there are many possible reasons other than oligomeric state that might explain the inability of Salmonella FliD to complement Pseudomonas.

We have acknowledged this by adding to the following sentence to the end of the paragraph describing these experiments, which reads: “Although there exist many possible reasons other than oligomeric state that could explain the inability of *Salmonella* FliD to functionally complement *Pseudomonas*, these data suggest that *Pseudomonas* flagella may prefer FliD proteins that adopt hexameric rather than pentameric states.”